# Functional Cache Grafting:
# Robust and Rapid Code-Policy Synthesis for Embodied Agents

**Saehun Chun** [1]  **Wonje Choi** [1]  **Sera Choi** [1]  **Sanghyun Ahn** [1]  **Honguk Woo** [1]

## Abstract

Code-writing large language models (CodeLLMs) generate executable code policies for embodied agents by translating natural language goals and environmental constraints into structured control programs. However, policy generation in open-domain embodied environments suffers from two fundamental limitations: (i) delayed decoding caused by repetitive prefill computation over long prompts, and (ii) limited robustness due to fully generative decoding, which often produces API mismatches, missing safety guards, and unstable control logic. To address these limitations, we present FCGRAFT, a Functional Cache Grafting framework. FCGRAFT maintains a library of function-level validated code skeletons and their associated prompt-level Transformer key–value (KV) caches, and synthesizes new policies by retrieving relevant functions and grafting their KV caches when a new task is provided. Given retrieved function caches, FCGRAFT performs cache grafting via stitching, which composes cached function segments into a composite policy, and patching, which locally adapts only the necessary code regions to satisfy task-specific parameters and constraints with minimal additional decoding. By eliminating redundant prefill computation, this approach reduces generation latency, while reusing validated control structures improves robustness over prompt-level caching methods RAGCache, achieving $18.31\%$ higher task success rate and $2.3\times$ faster policy synthesis.

## 1. Introduction

Embodied agents, including mobile and manipulator robots, operate in ever-changing environments and continuously encounter novel tasks. To act effectively, agents must ground natural language instructions in perception and transform them into action policies that can handle open-ended, dynamic tasks with responsiveness and adaptability. Recent advances in large language models with code-writing capabilities (CodeLLMs) have inspired the paradigm of Code-as-Policies (CaP), in which executable control code is generated from instructions using a predefined set of APIs (Liang et al., 2023; Vemprala et al., 2023; Huang et al., 2023a;b; Burns et al., 2024). By leveraging the generalization capabilities of CodeLLMs, the CaP paradigm complements task-specific policy learning and provides a more flexible means of control. However, its practicality remains limited in real-world deployments. Generating full code from scratch often produces unstable code with API mismatches, missing safety guards, and faulty control logic, directly leading to task failures. Furthermore, CaP prompts often include lengthy specifications and examples, and fully generative decoding requires repetitive prefill over thousands of tokens for every new instruction, resulting in generation latency that undermines responsiveness in time-critical settings.

Ensuring robust code generation while minimizing redundant computation is therefore critical for practical CaP deployment. Promising directions include memory-based approaches that reuse prior experiences, together with key-value (KV) caching mechanisms that minimize redundant attention computations. However, existing memory-based approaches in embodied agents primarily operate at the text level, offering only limited improvements in computational efficiency. Likewise, prior work on KV caching does not support function-level adaptation and is therefore inadequate for the dynamic and evolving nature of open-domain environments. To this end, we propose FCGRAFT, a Functional Cache Grafting framework that enhances CodeLLM-based robotic programming via function-level KV caching. FCGRAFT repurposes the native KV caching mechanism to enable function-level code reuse, forming the basis for cache-grafted code policy synthesis. Akin to the skill-based paradigm in robotics, where predefined skills, rep-

---

[1]Department of Computer Science and Engineering, Sungkyunkwan University, Suwon, Republic of Korea. Correspondence to: Honguk Woo <hwoo@skku.edu>.

*Proceedings of the $43^{rd}$ International Conference on Machine Learning*, Seoul, South Korea. PMLR 306, 2026. Copyright 2026 by the author(s).

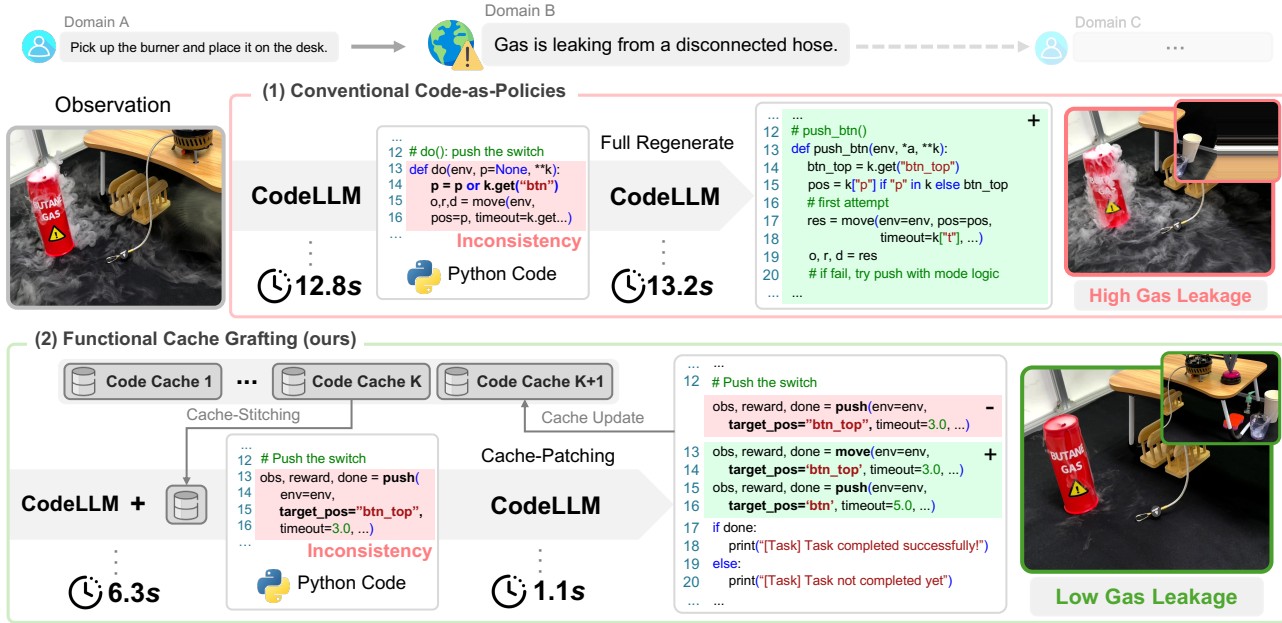

*Figure 1.* Illustration of FCGRAFT in an open-domain scenario involving gas management. (1) Conventional CaP incurs high latency from repetitive prefill and low robustness from fully generative decoding; delayed responses cause gas leakage, cascading into further disruptions. (2) FCGRAFT employs cache-grafting (cache-stitching and cache-patching) to eliminate redundant prefill and reuse validated control structures, enabling rapid and robust policy synthesis.

resented as action sequences, are combined and adapted to meet environment-specific constraints and task variations (Nachum et al., 2018; Nyga et al., 2018; Rana et al., 2023), our framework follows the principle of reusing prior building blocks. However, FCGRAFT focuses on efficient policy construction and improved responsiveness by reusing and composing cached KV states, enabling targeted modifications without full regeneration.

Figure 1 shows the core distinction between conventional CaP and FCGRAFT. While conventional CaP generates full control code from scratch for each instruction, FCGRAFT leverages cached functions to enable efficient and reliable code policy synthesis in open-domain environments. To effectively handle unpredictable situations, FCGRAFT performs cache-stitching and cache-patching not as independent techniques, but as two interdependent stages of a unified pipeline to address fundamentally different error types.

From earlier task successes, generated code policies are decomposed into function-level KV caches and stored in a two-tier code cache, where lightweight references are provided and complete implementations are retained. Building on cached functions, cache-stitching composes function calls directly through their KV states, eliminating redundant prefill computation while ensuring robustness by reusing verified control structures and API call patterns. Cache-patching then localizes error spans and generates only the corrected

portion, dramatically reducing decoding time compared to full regeneration. Consequently, FCGRAFT reduces CaP's overhead while improving robustness.

We evaluate FCGRAFT across diverse open-domain scenarios using embodied benchmarks, including ALFRED (Shridhar et al., 2020), TEACh (Padmakumar et al., 2022), and RLBench (James et al., 2020), as well as real-world robotic manipulation tasks. Experimental results demonstrate that FCGRAFT achieves a superior trade-off between robustness and latency compared to RAGCache (Jin et al., 2024), improving task success rates by 18.31% and reducing policy synthesis latency by 2.3× on average (Tables 1 & 2). These results highlight FCGRAFT's ability to deliver fast, reliable, and consistent policy synthesis, showing its practical advantages for embodied agents in open-domain environments.

Our contributions are summarized as follows:

- We present the FCGRAFT framework that improves CodeLLM-based robotic programming through function-level KV caching, reducing redundant prefill while achieving behavioral consistency in open-domain environments.

- We introduce cache-stitching and cache-patching as interdependent mechanisms: stitching eliminates structural errors and reduces prefill; patching corrects localized errors with minimal decoding.

- We show that FCGRAFT achieves the best trade-off be-

tween robustness and latency in diverse open-domain scenarios, including real-world robotic manipulation.

## 2. Related work

**Large language models for embodied control.** In embodied control, there is a growing trend of leveraging the reasoning capabilities of LLMs for task planning without additional finetuning (Huang et al., 2022b; Brohan et al., 2023; Song et al., 2023; Wang et al., 2023; Yao et al., 2023; Liang et al., 2024; Huang et al., 2023c; Zhou et al., 2023; Zhao et al., 2024; Huang et al., 2022a). Building on recent advances in the code-writing capabilities of LLMs (Chen et al., 2021; Nijkamp et al., 2022; Roziere et al., 2023; Hui et al., 2024; Guo et al., 2024; Zhu et al., 2024), the paradigm of LLM-based embodied control has shifted toward generating executable code as control policies (Liang et al., 2023; Huang et al., 2023b;a; Burns et al., 2024; Li et al., 2024; Mu et al., 2024; Vemprala et al., 2023; Singh et al., 2022). Beyond mapping instructions to predefined skills or action primitives, these frameworks prompt CodeLLMs to produce Python-like scripts that directly invoke perception and motor APIs, enabling embodied agents to perform motion-level control. In this work, we extend the CaP framework to improve robotic programming, overcoming the latency and inconsistency that arise from full regeneration of code policies in dynamic environments.

**Memory-based embodied agents.** Recent work has explored equipping embodied agents with memory to support long-horizon reasoning, task adaptation, and generalization by reusing validated experiences (Xu et al., 2025; Shinn et al., 2024; Li et al., 2025b; Kang et al., 2023; Wang et al., 2024; Kagaya et al., 2024). Such approaches maintain structured buffers of past experiences, such as observations, trajectories, and latent representations, to facilitate more informed decision-making. In the context of CaP, only a few studies have explored leveraging memory to enhance code writing capabilities (Sarch et al., 2023; 2024; Tziafas & Kasaei, 2024). However, these systems mainly focus on task generalization and adaptation, with less emphasis on real-time responsiveness. Unlike prior memory-based approaches that primarily rely on text-level representations for generalization, FCGRAFT introduces function-level KV caching tailored for function reuse, enabling rapid and reliable cache-augmented code policy synthesis.

**Key-value caching.** Recent works (Chan et al., 2025; Jin et al., 2024; Lu et al., 2024), extending retrieval-augmented generation, incorporate KV caching methods that accelerate generation by reusing precomputed attention states at the document level. While effective in reducing computational redundancy, these approaches are not well suited for dynamic or continually changing contexts. To handle such contexts, PromptCache (Gim et al., 2024) proposes a modular KV cache reuse strategy that composes multiple KV cache segments under changing prompt structures without being restricted to fixed prefix positions. Furthermore, methods such as CacheBlend (Yao et al., 2025), EPIC (Hu et al., 2024), and MPIC (Zhao et al., 2025) improve non-prefix KV reuse by selectively correcting boundary-level inter-segment attention dependencies instead of recomputing entire cached segments. In parallel, methods such as FIM (Bavarian et al., 2022), PIE (He et al., 2024), and EFIM (Guo et al., 2025) introduce infilling techniques for fill-in-the-middle (FIM), enabling new code lines to be inserted into cached sequences. Inspired by these advances, FCGRAFT adapts modular KV caching to embodied control, supporting cache-stitching for robust function-level code composition and cache-patching for efficient task-specific adaptation.

## 3. Problem formulation

We formulate the open-domain embodied task as a tuple $(\mathcal{D}, \mathcal{S}, \mathcal{A}, \mathcal{F})$, where $\mathcal{D}$ is the set of domains, $\mathcal{S}$ the state space, $\mathcal{A}$ the action space, and $\mathcal{F}$ the domain mapping function. Due to partial observability (Sutton & Barto, 2018), at each timestep $t$, the agent perceives an observation $o_t \in \mathcal{O}$ that provides incomplete information about the true state $s_t \in \mathcal{S}$. The function $\mathcal{F}(d) = \{\Omega_d, \mathcal{G}_d, \mathcal{P}_d\}$ (Hallak et al., 2015) maps each domain $d$ to its observation function $\Omega_d : \mathcal{S} \times \mathcal{A} \to \mathcal{O}$, a set of goal states $\mathcal{G}_d \subset \mathcal{S}$, and transition function $\mathcal{P}_d : \mathcal{S} \times \mathcal{A} \to \mathcal{S}$. The target goal state $g_d \in \mathcal{G}_d$ is defined by a natural language instruction set $\mathcal{T}_d$, where each instruction $\tau_d \sim \mathcal{T}_d$ specifies a corresponding goal condition. In open-domain settings, $\mathcal{D}$ is neither fixed nor known in advance, requiring the agent to continually adapt to unpredictable tasks and ever-changing environments (Choi et al., 2025). Instead of modeling the policy as a direct mapping from $(o_t, \tau_d)$ to $a_t$, we employ a CodeLLM $\pi_\theta$ to generate an executable code policy $\pi_{\text{code}}$ that specifies this mapping. Our objective is to optimize $\pi_\theta$ as follow:

$$\pi_\theta^* = \arg\max_{\pi_\theta} \mathbb{E}_{d \sim \mathcal{D}} \mathbb{E}_{\pi_{\text{code}} \sim \pi_\theta(\cdot | \tau_d, o_t)} \\ \left[ \text{SR}(\text{Exec}(\pi_{\text{code}}), g_d) - \eta \text{PSL}(\pi_\theta) + \mu \text{CSIM}(\pi_\theta) \right] \quad (1)$$

where the generated $\pi_{\text{code}}$ maximizes task success rate (SR) while minimizing policy synthesis latency (PSL), with code similarity (CSIM) as a regularizer for consistent behavior across tasks. Here, Exec denotes the trace of program execution in domain $d$, and $\eta$ and $\mu$ are weighting factors. We note that this objective is an idealized formulation intended to summarize the desired trade-off among task success, synthesis latency, and behavioral consistency, rather than a training loss optimized directly.

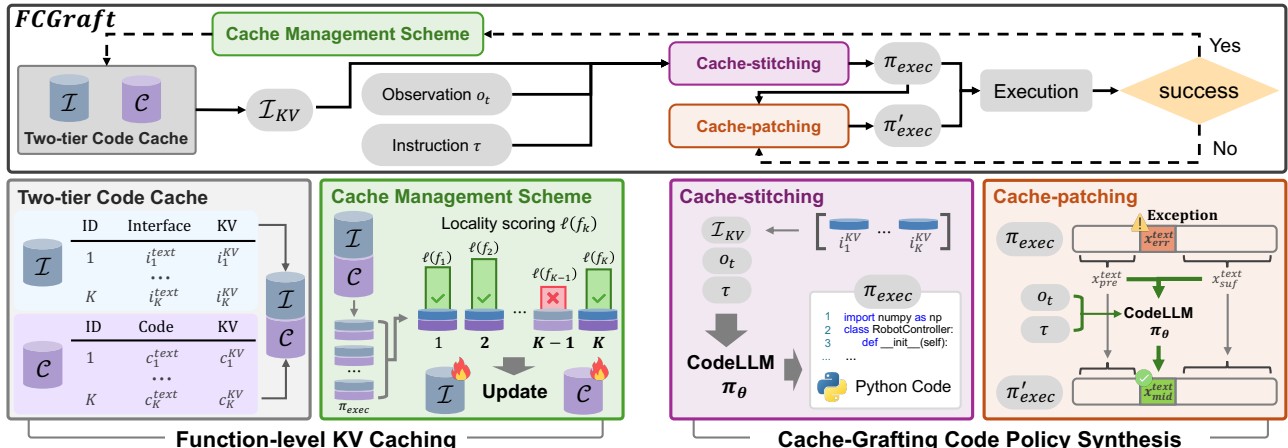

*Figure 2.* Overall architecture of FCGRAFT. **Top:** End-to-end robotic programming workflow. **Bottom:** Process of function-level KV caching and cache-grafting code policy synthesis.

## 4. FCGRAFT: Functional Cache Grafting

As illustrated in Figure 2, FCGRAFT is built on two core mechanisms: (i) function-level KV caching and (ii) cache-grafting code policy synthesis. In Section 4.1, previously generated and validated code policies are decomposed into callable functions and stored in a two-tier code cache, with each tier maintaining both textual and KV representations. The code cache organizes function-level entries into an interface tier and an implementation tier, facilitating lightweight code generation and in-place editing during code policy synthesis. To ensure reusability across open-domain tasks, our framework dynamically manages cached functions not only based on recent usage patterns but also through a semantic-aware strategy that accounts for functional diversity. In Section 4.2, new policies are synthesized through cache-stitching and cache-patching, two interdependent stages of a unified pipeline. Stitching composes pre-validated function segments, eliminating structural errors and reducing prefill. Patching then corrects remaining errors with minimal decoding. This separation enables FCGRAFT to achieve both robustness and efficiency.

### 4.1. Function-level KV caching

As illustrated in Figure 2, we implement a two-tier code cache $\mathcal{H}$ with a Function-Interface tier $\mathcal{I}$ and a Function-Code tier $\mathcal{C}$, both indexed by function identifier $f_k$.

$$\mathcal{H} = (\mathcal{I}, \mathcal{C}), \quad \mathcal{I} = \{(f_k, i_k^{\text{text}}, i_k^{\text{KV}})\}_{k=1}^K, \quad (2)$$
$$\mathcal{C} = \{(f_k, c_k^{\text{text}}, c_k^{\text{KV}})\}_{k=1}^K$$

The native KV caching mechanism is restructured into this two-tier hierarchical design, enabling efficient function reuse during inference. $\mathcal{I}$ maintains callable signatures along with semantic metadata abstracting full implementations to the function interfaces with KV states for lightweight cache-grafted code generation. When available, $\mathcal{C}$ preserves the corresponding validated implementations that are directly executable and editable in-place within KV states. To support continual reuse during runtime, each function entry is scored based on recency and invocation frequency, with consideration of conditional co-occurrence in execution traces and semantic functional diversity.

**Two-tier code cache.** The first tier, Function-Interface $\mathcal{I}$, stores entries containing function interfaces $i_k^{\text{text}}$ in text form, which define the function name and typed parameters, paired with their KV states $i_k^{\text{KV}}$. Each entry is modularized at the function level, allowing its $i_k^{\text{KV}}$ to be reinjected into the CodeLLM $\pi_\theta$ without recomputing cross-attention. This design follows the modular KV reuse principle that reusable segments can be composed without being tied to fixed prefix positions (Hu et al., 2024; Gim et al., 2024). By treating each function interface as an independent module, FCGRAFT reduces redundant attention computation and mitigates the *attention sink* problem when multiple cached functions are handled together. The second tier, Function-Code $\mathcal{C}$, stores validated implementations $c_k^{\text{text}}$ with their cached KV states $c_k^{\text{KV}}$, also indexed by $f_k$. These entries are excluded from direct injection into $\pi_\theta$ during generation to avoid redundant reasoning over complex implementations. Instead, they are linked at execution time to assemble the final program, while their KV states serve as prefix context when patching errors within cached functions.

**Cache management scheme.** After executing each task, all successfully invoked functions, including newly generated ones, are decomposed into $\mathcal{I}$ and $\mathcal{C}$ entries and stored in $\mathcal{H}$. To manage $\mathcal{H}$ under limited GPU memory, we assign

**Algorithm 1** Procedure of FCGRAFT: (A) task execution loop and (B) built-in methods

**(A) Task Execution Loop**

Agent FCGRAFT; Environment $env$

1:  $scenario\_done \leftarrow$ False
2:  **while** not $scenario\_done$ **do**
3:      $t \leftarrow 0; (o_t, \tau) \leftarrow env.\text{reset}()$
4:      $\pi_{\text{exec}} \leftarrow \text{FCGRAFT.stitch}(o_t, \tau)$
5:      $[a_0, a_1, \ldots] \leftarrow \text{Exec}(\pi_{\text{exec}})$
6:      info $\leftarrow \{$"is_done" : False$\}$
7:      **while** not info["is_done"] **do**
8:          **try**:
9:              $(o_{t+1}, \text{info}) \leftarrow env.\text{step}(a_t)$
10:         **except** Exception as $E$:
11:             $\pi_{\text{exec}} \leftarrow \text{FCGRAFT.patch}(o_{t+1}, \tau, E, \pi_{\text{exec}})$
12:             $[a_t, a_{t+1}, \ldots] \leftarrow \text{Exec}(\pi_{\text{exec}})$
13:             **continue**
14:         $o_t \leftarrow o_{t+1}; t \leftarrow t+1$
15:     **end while**
16:     **if** info["is_success"] **then**
17:         FCGRAFT.update($\pi_{\text{exec}}$)
18:     **end if**
19:     $scenario\_done \leftarrow$ info["scenario_done"]
20: **end while**

**(B) FCGRAFT Built-in Methods**

Two-tier code cache $\mathcal{H} = (\mathcal{I}, \mathcal{C})$

**def** stitch(self, $o_t, \tau$):
  # cache-stitching
  1:  Generate $\pi_{\text{code}}$ using Eq.(4)
  2:  Get $\pi_{\text{exec}}$ using Eq.(5)
  3:  $\pi_{\text{exec}} \leftarrow \text{self.patch}(o_t, \tau, E, \pi_{\text{exec}})$ **if** $E$ in $\pi_{\text{exec}}$
                              **else** $\pi_{\text{exec}}$
  4:  **return** $\pi_{\text{exec}}$

**def** patch(self, $o_t, \tau, E, \pi_{\text{exec}}$):
  # cache-patching
  1:  Split $\pi_{\text{exec}} = [x_{\text{pre}}^{\text{text}} \| x_{\text{err}}^{\text{text}} \| x_{\text{suf}}^{\text{text}}]$
  2:  $x_{\text{mid}}^{\text{text}} \leftarrow \pi_\theta(x_{\text{pre}}^{\text{KV}}, \text{CoT}(o_t, \tau, c_k^{\text{KV}}, E))$
  3:  $\pi_{\text{exec}}' \leftarrow [x_{\text{pre}}^{\text{text}} \| x_{\text{mid}}^{\text{text}} \| x_{\text{suf}}^{\text{text}}]$      *cf. Eq.(6)*
  4:  **return** $\pi_{\text{exec}}'$

**def** update(self, $\pi_{\text{exec}}$):
  # cache management scheme
  1:  Decompose $\pi_{\text{exec}}$ into $\mathcal{I}$- and $\mathcal{C}$-entries
  2:  Assign the locality score $\ell(f_k)$ using Eq.(3)
  3:  Update $\mathcal{H}$ under limited GPU memory
  4:  **return** 0

---

a locality score $\ell(f_k)$ to each function $f_k$, indicating reuse potential. Low-scored entries, particularly $c_k^{\text{KV}}$ in $\mathcal{C}$, are offloaded to DRAM, while high-scored entries are retained on the GPU to support fast revisit during code policy synthesis.

$$\ell(f_k) = (1 - \ell_{\text{curr}}(f_k)) \cdot (\alpha \cdot \ell_{\text{freq}}(f_k) \\ + \beta \cdot \sum_j \ell_{\text{asso}}(f_j \mid f_k) + \gamma \cdot \ell_{\text{sema}}(f_k)) + \ell_{\text{curr}}(f_k) \quad (3)$$

Here, $\ell_{\text{curr}}(f_k) \in \{0, 1\}$ is a binary indicator denoting whether $f_k$ was recently used, enforcing immediate retention. The composite score integrates: (1) usage frequency $\ell_{\text{freq}}$, (2) conditional association $\ell_{\text{asso}}$ with co-invoked functions, and (3) semantic relevance $\ell_{\text{sema}}$ based on perplexity (Jelinek et al., 1977). Each component is normalized to $[0, 1]$ and weighted by coefficients $\alpha$, $\beta$, and $\gamma$ satisfying $\alpha + \beta + \gamma = 1$. This scoring captures both short-term recency and long-term functional utility while promoting semantic diversity of cached functions. It enables the agent to maintain a compact yet adaptive code cache that remains responsive to the unpredictability of open-domain tasks (see Appendix Table 28 and Table 29 for detailed analysis).

### 4.2. Cache-grafting code policy synthesis

As shown in Figure 2, leveraging the two-tier code cache $\mathcal{H}$, FCGRAFT synthesizes new code policies through cache-stitching (using $\mathcal{I}$) and cache-patching (using $\mathcal{C}$). In cache-stitching, FCGRAFT generates a code policy $\pi_{\text{code}}$ by invoking cached functions via their KV states stored in the

Function-Interface tier $\mathcal{I}$. This allows executable policies to be assembled compositionally from cached functions. When task-specific revisions are required due to unexpected changes or execution errors, FCGRAFT enters cache-patching mode. It revisits $\pi_{\text{code}}$ along with corresponding implementations in the Function-Code tier $\mathcal{C}$, reusing prefix KV states and generating only the corrected span.

**Cache-stitching.** Cache-stitching connects cached function segments to compose new policies, bypassing redundant prefill. Crucially, stitching eliminates internal errors. Code errors fall into two categories: internal errors (incorrect API sequences, faulty control logic) within function implementations, and surface errors (mismatched parameters, incorrect variable values) that can be corrected without modifying the function body. By reusing pre-validated KV caches, stitching ensures that any errors in the synthesized policy are surface-level rather than internal, enabling effective patching downstream. Given current observation $o_t$, task instruction $\tau$, and cached interface KV states $i_k^{\text{KV}}$ in $\mathcal{I}$, the CodeLLM $\pi_\theta$ generates code policy $\pi_{\text{code}}$.

$$\pi_{\text{code}} \sim \pi_\theta(\cdot \mid o_t, \tau, \mathcal{I}_{\text{KV}}), \\ \mathcal{I}_{\text{KV}} = \{i_k^{\text{KV}} \mid (f_k, i_k^{\text{text}}, i_k^{\text{KV}}) \in \mathcal{I}\} \quad (4)$$

Here, $\mathcal{I}_{\text{KV}}$ denotes the set of interface KV states, which are concatenated and injected into $\pi_\theta$ as additional context during generation. Let $\text{Call}(\pi_{\text{code}})$ denote the set of function identifiers that appear as function calls in $\pi_{\text{code}}$. The final executable program $\pi_{\text{exec}}$ is constructed by linking

the cached implementations from the Function-Code tier $\mathcal{C}$. During this process, the KV states for $\pi_{\text{exec}}$ are computed and retained for potential use in cache-patching.

$$\pi_{\text{exec}} = \pi_{\text{code}} \| \{ c_k^{\text{text}} \mid i_k^{\text{text}} \in \text{Call}(\pi_{\text{code}}), \\ (f_k, i_k^{\text{text}}, i_k^{\text{KV}}) \in \mathcal{I}, (f_k, c_k^{\text{text}}, c_k^{\text{KV}}) \in \mathcal{C} \} \quad (5)$$

Here, $\|$ denotes the concatenation of the generated $\pi_{\text{code}}$ with the corresponding cached function $c_k^{\text{text}}$ from $\mathcal{C}$, prior to execution by the program executor.

**Cache-patching.** Cache-patching targets localized errors in the code produced by stitching, drastically reducing decoding time compared to full regeneration. The two mechanisms are thus interdependent: stitching separates invariant regions from adaptation points, and patching efficiently addresses the latter. Specifically, cache-patching is triggered when an exception $E$ is raised during execution, such as API mismatches or runtime errors. $\pi_{\text{exec}}$ is split into three parts: $\pi_{\text{exec}} = [x_{\text{pre}}^{\text{text}} \| x_{\text{err}}^{\text{text}} \| x_{\text{suf}}^{\text{text}}]$, where $x_{\text{pre}}^{\text{text}}$ and $x_{\text{suf}}^{\text{text}}$ denote the preserved prefix and suffix, and $x_{\text{err}}^{\text{text}}$ corresponds to the erroneous span responsible for $E$. Cache-patching reuses the KV states of the prefix $(x_{\text{pre}}^{\text{KV}})$ to generate only the middle span, guided by a CoT trace (Wei et al., 2022) that identifies the root cause of $E$ and is discarded before execution. The generated middle span is then concatenated with the suffix text $(x_{\text{suf}}^{\text{text}})$ to produce the corrected code.

$$\pi'_{\text{exec}} = \left[ x_{\text{pre}}^{\text{text}} \| x_{\text{mid}}^{\text{text}} = \pi_\theta(x_{\text{pre}}^{\text{KV}}, \text{CoT}(\cdot, c_k^{\text{KV}}, E)) \| x_{\text{suf}}^{\text{text}} \right] \quad (6)$$

By reusing $x_{\text{pre}}^{\text{KV}}$, cache-patching bypasses prefill computation for the prefix and focuses decoding only on the erroneous span. Algorithm 1 outlines the overall procedure of FCGRAFT.

## 5. Evaluation

### 5.1. Experimental setting

**Environments.** We evaluate FCGRAFT on widely used embodied benchmarks, including ALFRED (Shridhar et al., 2020), TEACh (Padmakumar et al., 2022), and RLBench (James et al., 2020), as well as real-world robotic manipulation. We construct 3 open-domain scenarios for each benchmark with increasing environmental dynamics: (1) *Open-Composition*, where tasks follow a curriculum-style progression and later tasks compose earlier ones; (2) *Open-Perturbation*, where object states (e.g., open vs. closed) may change unpredictably during execution; (3) *Open-Evolution*, where both observations and goal conditions vary.

**Tasks.** In each scenario, agents encounter a continual stream of tasks, categorized into simple tasks that require shallow logic and complex tasks that demand subtask composition or adaptive reasoning. In ALFRED, tasks such as

*move an item* are simple, while tasks such as *put items in correct places* require reasoning on spatial and temporal dependencies. In the real-world setting, we use complex tasks such as *organizing an office desk* and *preparing a cooking workstation*, which involve manipulating multiple objects with varying attributes and spatial constraints.

**Baselines.** We compare FCGRAFT against nine competitive baselines, categorized into three groups based on their strategy for leveraging the CodeLLM during code policy synthesis. (1) General CodeLLM-based programming methods such as **CaP** (Liang et al., 2023) and **SCoT** (Li et al., 2025a) directly generate code policies from natural language instructions. (2) Memory-based approaches such as **HELPER** (Sarch et al., 2023), **LRLL** (Tziafas & Kasaei, 2024), and **PromptBook** (Arenas et al., 2024) retrieve textual programming artifacts as additional inputs to the CodeLLM. (3) KV caching methods such as **CAG** (Chan et al., 2025), **RAGCache** (Jin et al., 2024), **PromptCache** (Gim et al., 2024), and **EPIC** (Hu et al., 2024) accelerate inference by reusing cached attention states. To ensure a fair comparison, all baselines are given the same prior information derived from external oracle policies for basic functions that are entirely unrelated to any task in the benchmarks. This prior information is supplied in the format appropriate for each method: prompt-based baselines receive it as prompts, memory-based baselines receive it as memory entries, and both the KV cache baselines and FCGRAFT receive it in KV cache form to establish an equivalent initialization.

**Metrics.** We evaluate performance across four complementary dimensions: (1) Task accuracy, measured by task success rate (SR) and goal condition success rate (GC) (Shridhar et al., 2020). (2) Computational efficiency, measured by policy synthesis latency (PSL) in seconds and number of generated tokens (NGT). (3) Behavioral consistency, measured by code similarity (CSIM) (Chon et al., 2024), forward transfer (FWT), and backward transfer (BWT) (Lopez-Paz & Ranzato, 2017). (4) Memory efficiency, measured by cache hit rate (HR) and GPU memory usage (MU).

**Implementation.** All experiments are conducted in Python 3.9 using Qwen2.5-Coder-14B (Hui et al., 2024) as the default CodeLLM, accessed via HuggingFace (Wolf et al., 2019), unless otherwise specified. For fair comparison, all baselines use the same CodeLLM configuration and run on off-the-shelf NVIDIA RTX 4090 GPUs. Experimental details are provided in Appendix A and Appendix B.

*Table 1.* Performance on open-domain embodied tasks across simulation benchmarks. Results are averaged over three seeds, with standard deviations indicating consistency across runs.

| **ALFRED** | *Open-Composition* | | | *Open-Perturbation* | | | *Open-Evolution* | | |
|---|---|---|---|---|---|---|---|---|---|
| Methods | SR (↑) | PSL (↓) | Rank (↓) | SR (↑) | PSL (↓) | Rank (↓) | SR (↑) | PSL (↓) | Rank (↓) |
| CaP | 46.48±1.87 | 16.21±1.03 | 8 (0.53) | 42.23±0.12 | 16.15±0.23 | 8 (0.53) | 37.22±1.72 | 16.28±1.20 | 8 (0.50) |
| SCoT | 47.93±0.87 | 30.02±3.38 | 10 (0.27) | 43.46±1.06 | 32.39±1.12 | 10 (0.23) | 39.00±0.98 | 32.04±2.10 | 10 (0.22) |
| HELPER | 53.83±0.39 | 16.82±1.20 | 6 (0.55) | 52.49±0.24 | 15.94±0.69 | 3 (0.58) | 47.75±1.08 | 16.42±0.92 | 3 (0.55) |
| LRLL | 56.28±3.20 | 16.02±2.02 | 2 (0.58) | 54.70±0.87 | 16.40±1.24 | 2 (0.59) | 51.60±0.30 | 16.63±1.03 | 2 (0.57) |
| PromptBook | 53.94±3.96 | 31.34±1.82 | 9 (0.27) | 52.63±1.09 | 33.29±1.52 | 9 (0.26) | 50.49±0.51 | 33.49±2.07 | 9 (0.25) |
| CAG | 38.24±0.04 | 12.12±0.62 | 3 (0.57) | 36.03±0.32 | 12.38±0.92 | 4 (0.57) | 32.18±1.12 | 12.39±0.92 | 4 (0.55) |
| RAGCache | 39.29±1.70 | 13.48±0.93 | 7 (0.55) | 36.33±1.02 | 13.15±0.12 | 6 (0.56) | 33.83±1.53 | 13.02±1.15 | 5 (0.55) |
| EPIC | 43.38±1.02 | 14.02±1.04 | 5 (0.56) | 37.08±1.02 | 13.74±0.69 | 7 (0.55) | 34.23±0.23 | 13.49±1.14 | 7 (0.54) |
| PromptCache | 40.37±0.37 | 12.82±0.86 | 4 (0.56) | 36.92±2.22 | 12.93±1.48 | 5 (0.56) | 34.14±0.82 | 13.18±1.15 | 6 (0.54) |
| FCGRAFT (ours) | **61.58**±1.86 | **5.82**±0.57 | **1 (0.81)** | **57.23**±0.23 | **6.32**±0.51 | **1 (0.79)** | **55.89**±0.85 | **6.29**±1.25 | **1 (0.78)** |

| **RLBench** | *Open-Composition* | | | *Open-Perturbation* | | | *Open-Evolution* | | |
|---|---|---|---|---|---|---|---|---|---|
| Methods | SR (↑) | PSL (↓) | Rank (↓) | SR (↑) | PSL (↓) | Rank (↓) | SR (↑) | PSL (↓) | Rank (↓) |
| CaP | 31.23±3.32 | 11.39±0.53 | 8 (0.45) | 30.46±0.02 | 11.23±0.93 | 7 (0.46) | 26.84±1.20 | 11.23±0.24 | 7 (0.48) |
| SCoT | 32.32±2.22 | 22.03±2.03 | 9 (0.20) | 30.32±0.28 | 21.37±1.84 | 9 (0.22) | 27.11±0.37 | 22.83±1.42 | 9 (0.21) |
| HELPER | 35.42±1.32 | 11.48±0.23 | 7 (0.47) | 34.84±0.03 | 11.38±0.06 | 6 (0.48) | 31.12±1.08 | 12.16±1.42 | 8 (0.48) |
| LRLL | 36.11±0.00 | 10.83±0.82 | 4 (0.49) | 33.82±0.02 | 12.01±0.11 | 8 (0.46) | 32.45±0.94 | 12.17±2.01 | 6 (0.48) |
| PromptBook | 35.93±1.12 | 23.49±0.63 | 10 (0.18) | 33.03±0.67 | 24.03±3.89 | 10 (0.17) | 31.65±3.32 | 25.93±2.09 | 10 (0.16) |
| CAG | 26.37±2.03 | 8.53±0.83 | 2 (0.50) | 24.89±1.21 | 8.39±0.37 | 2 (0.50) | 22.82±0.73 | 8.08±1.91 | 2 (0.53) |
| RAGCache | 27.26±1.08 | 9.09±0.63 | 6 (0.49) | 27.03±1.04 | 9.28±0.66 | 5 (0.49) | 23.79±1.20 | 9.02±0.18 | 5 (0.51) |
| EPIC | 29.24±2.88 | 9.48±1.20 | 5 (0.49) | 29.04±0.83 | 9.48±0.32 | 4 (0.50) | 25.85±1.25 | 9.28±0.47 | 3 (0.52) |
| PromptCache | 28.33±1.73 | 8.99±0.39 | 3 (0.50) | 27.41±0.81 | 8.97±0.35 | 3 (0.50) | 25.03±0.85 | 9.12±0.65 | 4 (0.52) |
| FCGRAFT (ours) | **45.91**±4.37 | **3.24**±0.65 | **1 (0.73)** | **42.82**±2.64 | **3.47**±0.16 | **1 (0.71)** | **33.98**±1.27 | **4.39**±0.43 | **1 (0.67)** |

## 5.2. Main Result

**Benchmarks.** Table 1 presents a comparison between FCGRAFT and 9 competitive baselines for code policy synthesis in open-domain tasks, evaluated on 3 benchmarks, each with 3 open-domain scenarios. Results on TEACh are provided in Appendix 14. Across all benchmarks, FCGRAFT consistently outperforms the baselines, achieving the best trade-off between SR and PSL. On average, it achieves a $18.31\%$ higher SR and $2.3\times$ reduction in PSL compared to RAGCache. This improvement in SR is not merely a byproduct of reduced PSL. By reusing function-level KV states associated with previously validated interfaces and implementations, FCGRAFT biases policy synthesis toward reusable program structures rather than unconstrained generation. In the synthesized policies, this reuse helps preserve validated API-call sequences and control-flow templates, avoiding failures such as incompatible primitive compositions, missing condition checks, and inconsistent branches. As a result, function-level KV reuse improves both synthesis efficiency and policy robustness.

On ALFRED, FCGRAFT achieves a $4.04\%$ higher SR and a $2.67\times$ reduction in PSL compared to LRLL, the second-best baseline in Rank. On TEACh, which involves interactive tasks and frequent instruction changes, FCGRAFT shows similar improvements (Appendix 14). These results highlight the effectiveness of cache-stitching for robust code generation and cache-patching for efficient adaptation. Com-

pared to CaP, cache-stitching produces more reliable code by reusing validated functions, and cache-patching enables efficient corrections, together increasing function retention in the code cache. This accumulation creates a positive feedback loop: more retained functions improve stitching quality, which in turn simplifies patching for future tasks.

On RLBench, which emphasizes low-level manipulation skills and fine-grained control, FCGRAFT achieves a $16.21\%$ higher SR and a $2.25\times$ reduction in PSL compared to CAG. Unlike ALFRED and TEACh, RLBench tasks often involve loop structures that allow compact code representations, leading to generally lower PSL across all methods. The requirement for precise control makes RL-Bench particularly suitable for showcasing the strength of FCGRAFT's cache-patching, which enables fine-grained and efficient code-level adaptation. Specifically, in *Open-Composition*, FCGRAFT leverages cache-stitching followed by cache-patching to adjust function parameters or update variable values, resulting in improved SR despite increasing task complexity. In *Open-Perturbation*, cache-patching enables FCGRAFT to efficiently respond to changing object states by handling API-level exceptions triggered when actions fail, maintaining higher SR and faster PSL. In *Open-Evolution*, FCGRAFT adapts quickly to dynamic changes in both observations and goals by replacing initially selected functions with alternatives better suited to the environmental context, outperforming baselines in both SR and PSL.

*Table 2.* Performance on open-domain embodied tasks in real-world robotic manipulation.

| Real-world | Office Desk Rearrangement | | | Cooking Workstation Preparation | | |
|---|---|---|---|---|---|---|
| Methods | SR ($\uparrow$) | PSL ($\downarrow$) | Rank ($\downarrow$) | SR ($\uparrow$) | PSL ($\downarrow$) | Rank ($\downarrow$) |
| CaP | 33.33$\pm$0.00 | 11.94$\pm$1.17 | 4 (0.19) | 51.85$\pm$6.42 | 13.07$\pm$1.09 | 4 (0.26) |
| LRLL | 55.56$\pm$11.11 | 12.31$\pm$0.74 | 3 (0.28) | 55.56$\pm$0.00 | 12.65$\pm$1.14 | 3 (0.30) |
| RAGCache | 44.44$\pm$22.22 | 9.08$\pm$0.74 | 2 (0.41) | 37.04$\pm$6.42 | 9.63$\pm$1.01 | 2 (0.35) |
| FCGRAFT (ours) | **77.78**$\pm$11.11 | **3.80**$\pm$0.31 | **1 (0.89)** | **81.48**$\pm$12.83 | **2.85**$\pm$0.54 | **1 (0.91)** |

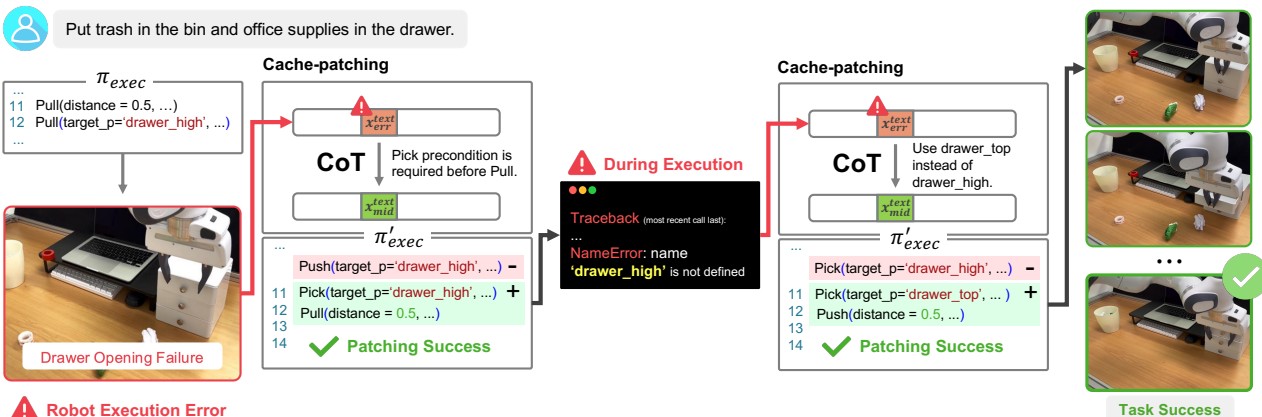

*Figure 3.* Qualitative examples of FCGRAFT's operation in real-world robotic manipulation.

**Robot tests.** In Table 2, we test FCGRAFT on real-world manipulation with a 7-DoF Franka Emika Research 3, evaluating transferability of the code cache from simulation to the physical robot. To build the code cache, the agent first solves simpler tasks in RLBench, then transfers to open-domain tasks involving mechanical failures and safety hazards. These include scenarios such as an office desk with diverse objects and complex spatial relations not seen in RLBench, and a cooking workstation where gas hoses disconnect mid-operation, requiring the agent to synthesize and deploy corrective code under time pressure. In both the office desk and cooking workstation scenarios, FCGRAFT outperforms all baselines, achieving 37.04% higher SR than CaP, reducing PSL by 2.88$\times$ compared to RAGCache, and showing reliable performance across 9 trials. Figure 3 complements Table 2 by demonstrating FCGRAFT's practicality through real-world examples where cache-stitching assembles policies by reusing cached functions, and cache-patching resolves execution-time errors. Specifically, when a runtime exception occurs during execution, the error is localized and the program is split into prefix, error, and suffix, after which FCGRAFT regenerates only the corrected span and re-executes the repaired code (e.g., replacing an inconsistent object-region parameter such as drawer_high with drawer_top).

## 5.3. Analysis and Ablation

**Analysis on behavior consistency.** Table 3 shows snapshot evaluations of the code cache at the *Initial*, *Middle*, and *Final* phases during a continual task stream of 27 tasks. At each phase, the entire task set is re-evaluated while preserving the current state of the code cache to assess the behavioral consistency of the accumulated code, reporting CSIM, FWT, BWT, and SR. The results show that FCGRAFT improves SR while maintaining consistent code structure across tasks; it achieves positive FWT without forgetting (non-negative BWT). These metrics reflect the interdependent roles of stitching and patching: high CSIM indicates that stitching preserves structural consistency by reusing validated function segments, while positive FWT shows that previously cached functions transfer effectively to new tasks. The non-negative BWT confirms that patching's localized corrections do not disrupt existing code structure. This separation, where stitching secures reusable structure and patching enabling non-destructive adaptation, is key to achieving continual improvement without forgetting.

**Ablation on model choice.** Table 4 reports the performance of FCGRAFT applied with four different LLM families, contrasting CodeLLMs with general-purpose LLMs. The results show that FCGRAFT equipped with CodeLLMs consistently outperforms its counterparts based on the LLMs, as well as CaP. On average, it achieves a 4.67%

*Table 3.* Analysis on behavioral and code-level consistency across continual task phases.

| RLBench | *Initial* (after task IDs 1–9) | | | | *Middle* (after task IDs 10–18) | | | | *Final* (after task IDs 19–27) | | | |
|---|---|---|---|---|---|---|---|---|---|---|---|---|
| Method | SR ($\uparrow$) | FWT | BWT | CSIM | SR ($\uparrow$) | FWT | BWT | CSIM | SR ($\uparrow$) | FWT | BWT | CSIM |
| CaP | 33.33 | 0.00 | 0.00 | 57.87±28.03 | 27.78 | 0.00 | 0.00 | 69.06±28.93 | 37.33 | – | 0.00 | 63.44±27.81 |
| LRLL | 33.33 | -6.25 | 5.56 | 72.17±14.84 | 33.33 | 0.00 | 5.56 | 52.87±21.88 | 40.00 | – | 4.00 | 48.25±24.98 |
| RAGCache | 33.33 | 0.00 | 0.00 | 34.12±1.36 | 33.33 | 0.00 | 0.00 | 27.77±8.49 | 36.44 | – | -2.00 | 35.74±13.69 |
| FCGRAFT (ours) | **44.44** | 2.08 | 0.00 | 65.63±5.03 | **38.89** | 4.76 | 0.00 | 57.58±11.87 | **46.00** | – | 2.00 | 55.43±8.98 |

higher SR, a $1.08\times$ reduction in PSL, and an $8.31\%$ higher HR compared to variants using LLMs, while maintaining a more favorable trade-off than CaP.

*Table 4.* Ablation on model choice.

| Family | Method | CodeLLM | SR ($\uparrow$) | PSL ($\downarrow$) | HR ($\uparrow$) |
|---|---|---|---|---|---|
| QWEN2.5 | FCGRAFT | ✔ | 29.62±0.01 | 2.89±0.41 | 58.63±1.28 |
| | FCGRAFT | ✘ | 26.38±0.23 | 3.12±0.43 | 47.12±3.94 |
| | CaP | ✔ | 23.63±0.16 | 7.94±0.47 | - |
| GEMMA | FCGRAFT | ✔ | 28.83±0.05 | 2.84±0.86 | 52.46±3.28 |
| | FCGRAFT | ✘ | 26.39±0.03 | 3.05±0.13 | 41.67±0.05 |
| | CaP | ✔ | 22.22±0.02 | 8.72±0.49 | - |
| LLAMA2 | FCGRAFT | ✔ | 31.12±1.32 | 3.32±0.09 | 57.91±4.27 |
| | FCGRAFT | ✘ | 25.33±1.07 | 3.46±0.26 | 52.85±1.29 |
| | CaP | ✔ | 17.83±2.46 | 11.48±0.23 | - |
| DEEPSEEK | FCGRAFT | ✔ | 30.23±5.71 | 2.33±0.34 | 67.23±3.28 |
| | FCGRAFT | ✘ | 23.04±3.21 | 2.64±0.86 | 61.34±3.42 |
| | CaP | ✔ | 18.98±0.41 | 13.07±1.13 | - |

**Ablation on FCGRAFT.** Table 5 reports the contribution of each component of FCGRAFT to overall performance. The ablation covers the two-tier code cache $\mathcal{H} = \{\mathcal{I}, \mathcal{C}\}$, the semantic locality term $\ell_{\text{sema}}$, and the policy-synthesis operation. For the code cache structure, removing the Function-Interface tier $\mathcal{I}$ (w/o $\mathcal{I}$), the Function-Code tier $\mathcal{C}$ (w/o $\mathcal{C}$), or the entire code cache $\mathcal{H}$ (w/o. $\mathcal{H}$) leads to a substantial drop in SR, confirming that the hierarchical design is crucial for improving robustness and reducing latency. Removing the semantic locality term in Eq. 3 (w/o $\ell_{\text{sema}}$) lowers HR and SR, showing that semantic diversity induced by $\ell_{\text{sema}}$ helps retain rare but useful functions that frequency- or association-based locality alone may overlook. Replacement of semantic scoring with retrieval-based cache selection ($\rightarrow$ Retrieval) reduces HR and increases PSL, indicating that retrieval-based matching is less effective at selecting reusable function caches in our setting. Replacement of localized cache-patching with full regeneration ($\rightarrow$ Regeneration) substantially increases PSL and reduces SR, showing that preserving validated surrounding code while repairing only the faulty span is important for both efficiency and robustness.

*Table 5.* Ablation on FCGRAFT. 'w/o' denotes removal of a component, and '$\rightarrow$' indicates replacement by another operation.

| Methods | SR ($\uparrow$) | PSL ($\downarrow$) | HR ($\uparrow$) |
|---|---|---|---|
| FCGRAFT | 45.91±4.37 | 3.24±0.65 | 73.65±2.15 |
| w/o $\mathcal{I}$ | 38.64±1.55 | 4.12±0.12 | 66.78±3.12 |
| w/o $\mathcal{C}$ | 34.37±0.52 | 3.33±0.23 | 67.56±3.51 |
| w/o $\mathcal{H}$ | 34.64±1.98 | 4.25±1.27 | - |
| w/o $\ell_{\text{sema}}$ | 35.02±1.76 | 3.74±0.29 | 48.61±1.96 |
| $\rightarrow$ Retrieval | 41.92±0.72 | 4.42±0.48 | 43.06±2.45 |
| $\rightarrow$ Regeneration | 35.77±1.76 | 6.89±0.20 | 71.39±2.83 |

grafted code policy synthesis. FCGRAFT stores generated code policies as function-level KV caches, enabling cache-stitching to reuse validated functions and reduce prefill, and cache-patching for efficient localized corrections. This interdependent design enables robustness and efficiency. Extensive experiments, including real-world manipulation, demonstrate the robustness and efficiency of FCGRAFT across tasks, highlighting that cache-based synthesis enables agents to generate and adapt code policies, a key step toward scalable and general-purpose embodied intelligence.

## 6.1. Limitation and Future Work

Despite its strengths, FCGRAFT maintains an isolated code cache for each agent, which may limit scalability in dynamic multi-agent cooperative settings. We plan to investigate code cache sharing across agents to enable multi-agent collaboration and to incorporate learning-based cache eviction and compression policies to optimize cache utilization.

## 6. Conclusion

We presented FCGRAFT, a Functional Cache Grafting framework that improves CodeLLM-based robotic programming through function-level KV caching and cache-

## Acknowledgement

This work was supported by Institute of Information & communications Technology Planning & Evaluation (IITP) grant funded by the Korea government (MSIT), (RS-2022-II220043, Adaptive Personality for Intelligent Agents, RS-2022-II221045, Self-directed multi-modal Intelligence for solving unknown, open domain problems, RS-2025-02218768, Accelerated Insight Reasoning via Continual Learning, RS-2025-25442569, AI Star Fellowship Support Program (Sungkyunkwan Univ.), RS-2026-25543726, Development of Leading Talent in Medical Domain-Specific Generative AI, RS-2026-25528384, Resource-Intensive AI Technologies Based on Sustainable GPU Integrated Platforms, RS-2019-II190421, Artificial Intelligence Graduate School Program (Sungkyunkwan University)), the National Research Foundation of Korea (NRF) grant funded by the Korea government (MSIT) (No. RS-2026-25474409), IITP-ITRC (Information Technology Research Center) grant funded by the Korea government (MSIT) (IITP-2025-RS-2024-00437633, 10%), IITP-ICT Creative Consilience Program grant funded by the Korea government (MSIT) (IITP-2026-RS-2020-II201821, 10%), the AI Computing Infrastructure Enhancement (GPU Rental Support) User Support Program funded by the Ministry of Science and ICT (MSIT), Republic of Korea (No. RQT-25-120157), and by Samsung Electronics Co., Ltd.

## Impact Statement

This paper presents work whose goal is to advance the field of Machine Learning. There are many potential societal consequences of our work, none of which we feel must be specifically highlighted here.

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

# A. Experiment setting

## A.1. RLBench

RLBench (James et al., 2020) is a large-scale benchmark and simulation environment for robotic manipulation, built around the Franka Emika Panda 7-DoF arm. Each task is defined as a goal-conditioned manipulation problem and can be executed in a photorealistic, interactive tabletop environment. As shown in Figure 4, the benchmark provides 100 hand-designed tasks ranging from simple primitives such as reaching and pushing to long-horizon activities such as opening an oven and placing a tray inside. Each task consists of one or more variations, and infinitely many episodes can be generated by randomizing object positions, colors, and shapes. Observations include RGB, depth, and segmentation masks from multiple cameras (stereo shoulder-mounted and wrist-mounted), along with proprioceptive states such as joint angles, velocities, torques, and end-effector pose. This diversity and realism make RLBench a suitable testbed for evaluating agents that require precise low-level control, visuomotor reasoning, and generalization across task variations.

**Environment.** Building on RLBench, we design three evaluation scenarios that reflect the unpredictability and uncertainty of open-domain environments. We evaluate on 20 benchmark tasks, each with variations, resulting in 40 task instances.

(1) In *open-composition*, tasks are presented in streams of gradually increasing difficulty, defined by the number of high-level primitive calls and whether the agent must reason about object geometry (e.g., size or spatial layout). For example, an agent may progress from simple move actions, to pick or pick-and-place, and eventually to more complex operations such as multiple pick-and-place or precise insert. This setting evaluates whether skills from earlier tasks can be reused and combined to solve more complex configurations, reflecting the compositional demands of open-domain settings.

(2) In *open-perturbation*, task descriptions are perturbed to mimic observation-level noise, such as referring to absent objects or omitting existing ones. For instance, the agent may encounter a description like *"There are two cups on the table"* when only one exists, or *"There is a button on the table"* when several are actually present. This scenario evaluates robustness under description-level uncertainty, requiring the agent to reconcile mismatches between the description and the actual scene while still pursuing the intended task goal.

(3) In *open-evolution*, task streams vary unpredictably in both difficulty and observation quality. Unlike *open-composition*, where complexity grows gradually, and *open-perturbation*, where mismatches arise within tasks, this scenario mixes tasks of all difficulty levels while also introducing perturbations at random. An easy instruction may be followed by a difficult one with altered observations, or vice versa. The aim is to test whether agents remain stable and effective across irregular, noisy task sequences, mirroring the uneven and unstructured nature of open-domain environments.

Since RLBench is goal-conditioned by design, we checked both success metrics. In practice, the goal-conditioned success rate differed only marginally from the standard success rate across all of our runs. For this reason, we report only the standard success rate in the tables.

**Task.** RLBench tasks are executed through a library of high-level primitive APIs that we built on top of the simulator's low-level interfaces (see Table 6). For clarity, we group them into three functional categories. These include object-centric manipulation (e.g., pick, place, push), object-referenced motion (e.g., move or align to a quaternion), and robot-internal control (e.g., open or close the gripper). Each primitive follows a structured signature with object- or pose-specific arguments, plus optional parameters for fine-grained control (e.g., offsets or approach axes). This modular design promotes interpretable and reusable skill calls, supporting compositional and generalizable policy construction.

We categorize RLBench tasks by the complexity of the required skill sequence. Easy tasks involve at most two primitive calls, such as *"Reach target"* or *"Pick up cup"*. Medium tasks typically require three or more calls, for example *"Put rubbish in bin"* or *"Unplug charger"*. Hard tasks involve multi-object goals or spatial constraints, such as *"Put all groceries in cupboard"* or *"Take usb out of computer"*. This hierarchy shows how simpler skills can be combined to solve harder tasks, providing a principled way to evaluate FCGRAFT's generalization to diverse objects and scenes.

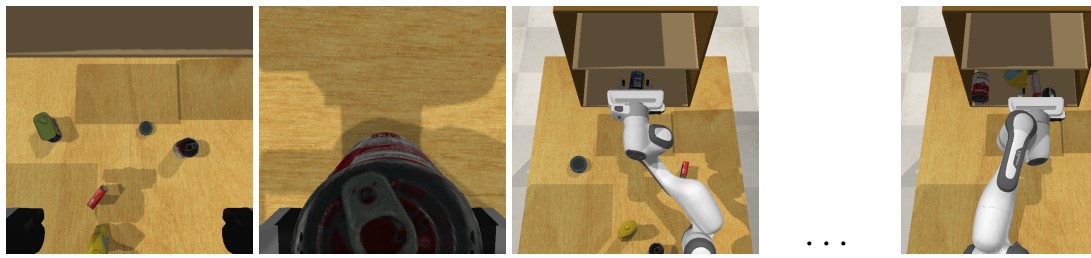

(a) Example of *"Put all groceries in cupboard"* with wrist view and overhead view

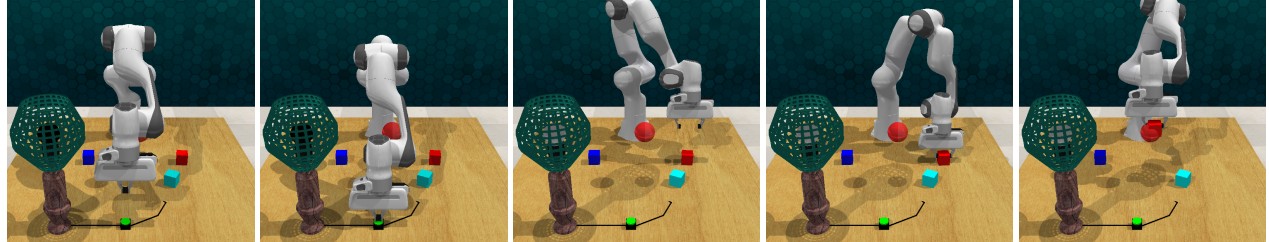

(b) Example of *"Lamp on and pick and lift the red block"* with front view

*Figure 4.* Environment examples set of RLBench.

*Table 6.* Instructions and executable APIs in RLBench.

|  | Template | Example |
|---|---|---|
| Instruction | Move | *"Reach the red sphere."* |
|  | Pick & Lift | *"Grasp the red cup and lift it."* |
|  | Pick & Place | *"Pick up the crackers and place them in the cupboard."* |
|  | Pick & Move | *"Take the charger out of the wall."* |
|  | Press | *"Press the maroon button."* |
|  | Push | *"Slide the block onto the target."* |
| API | Pick [Object] | `run_action(pick, 'crackers', offset=[0.0, 0.0, 0.2], approach_axis='z')` |
|  | Place [Receptacle Object] | `run_action(place, 'basket_ball_hoop', offset=[0.0, 0.0, 0.3], approach_dist=0.05)` |
|  | Move [Object] | `run_action(move, 'block', offset=[0, -0.1, 0])` |
|  | Push [Object] | `run_action(push, 'target_button', offset=[0.0, 0.0, -0.03])` |
|  | Align To Quaternion [Object] | `run_action(align_to_quaternion, 'usb', align_dir='parallel')` |
|  | Open Gripper | `run_action(open_gripper)` |
|  | Close Gripper | `run_action(close_gripper)` |

## A.2. ALFRED

ALFRED (Shridhar et al., 2020) is a large-scale benchmark for embodied AI that integrates vision-and-language navigation with manipulation-based rearrangement tasks. Each task is specified in natural language and requires agents to execute household activities in photorealistic 3D environments. As shown in Figure 5, the benchmark spans 120 indoor scenes (e.g., kitchens, living rooms), containing 58 manipulable object types (e.g., apple, phone) and 26 receptacle types (e.g., fridges, cabinets). From these components, 2685 unique task configurations are generated by combining one of seven instruction templates (e.g., pick-and-place, clean, heat) with scene and object variations. Examples of each template are provided in Figure 7. This diversity makes ALFRED a particularly suitable benchmark for evaluating agents on hierarchical reasoning, multi-step planning, and generalization across varied contexts.

**Environment.** To better emulate open-domain deployment, we design three long-horizon evaluation scenarios on top of ALFRED. (1) In *open-composition*, tasks are organized into a curriculum-like sequence where instruction types gradually increase in difficulty. The sequence begins with simple pick-and-place tasks that only require moving a single object to a receptacle, and then progresses to perception-oriented instructions such as examine. Subsequent stages introduce state-changing operations, including clean-and-place, heat-and-place, and cool-and-place, which require reasoning about object affordances and environment dynamics. Finally, the curriculum culminates in multi-object instructions such as pick-two-objects-and-place, which require hierarchical planning and the compositional reuse of previously synthesized policies. (2) In *open-perturbation*, tasks are subject to observation-level dynamics in which object states may change unpredictably during execution. For example, a cabinet that was initially open may become closed, or a light that was

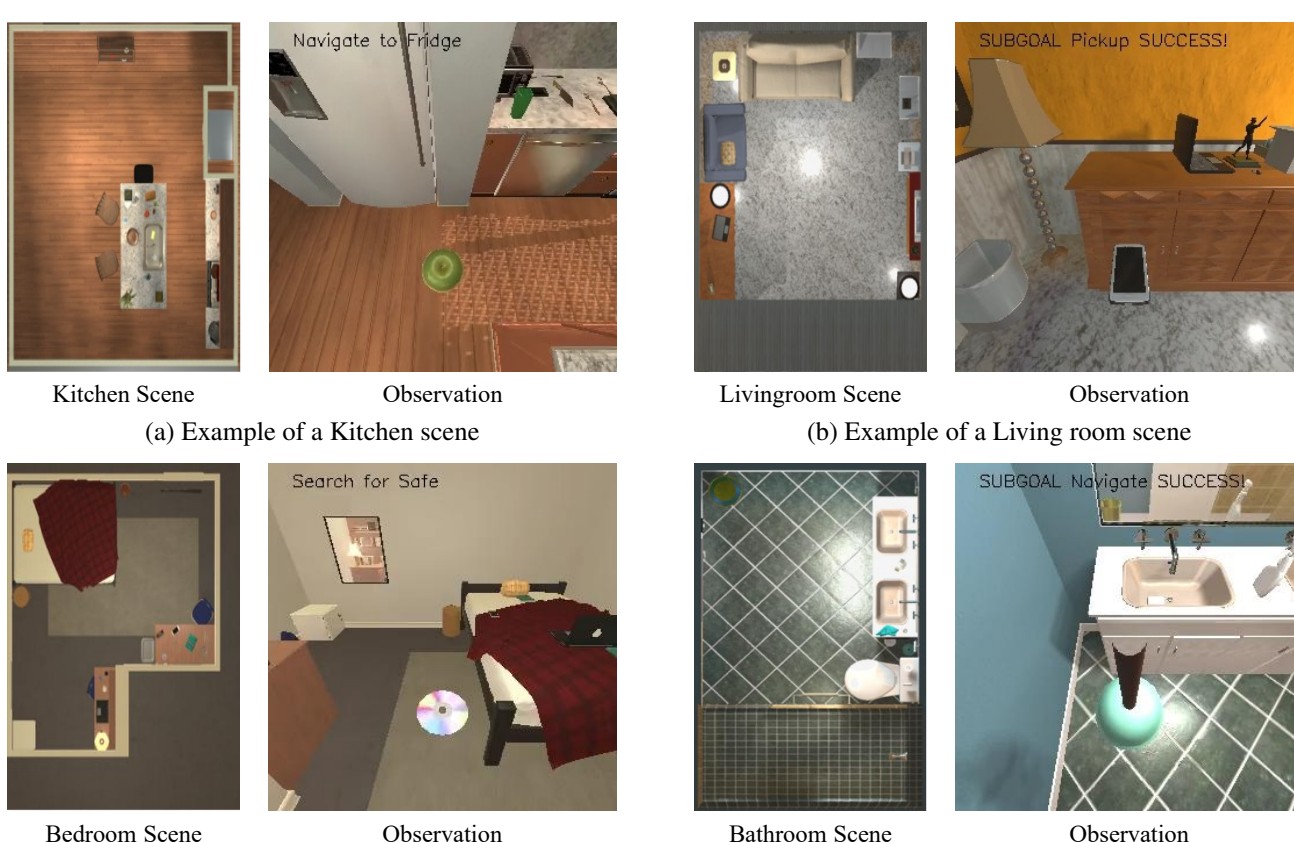

(a) Example of a Kitchen scene            (b) Example of a Living room scene

(c) Example of a Bedroom scene            (d) Example of a Bathroom scene

*Figure 5.* Examples of four scene types: kitchen, living room, bedroom, and bathroom.

turned on may switch off. Such perturbations require the agent to detect inconsistencies between expected and observed states, update its internal representation, and adapt its execution accordingly. This scenario evaluates robustness against environmental uncertainty while still preserving the original task goals. (3) In *open-evolution*, complexity is further increased by introducing changes not only in observations but also in goal conditions. For instance, the agent may be instructed to place a heated object on a plate, but midway through execution the target receptacle changes to a shelf. In this case, the agent must revise its plan, resynthesize or repair code policy in real time, and adapt its behavior to the updated objective. This scenario directly measures the agent's capacity for flexible replanning and generalization under evolving task specifications.

**Task.** ALFRED tasks are grounded in a library of action primitives (APIs) that enable interaction with objects and receptacles in the scene. As summarized in Table 7, these APIs cover both basic object manipulation (e.g., pickup, put, open, close) and state-altering operations (e.g., clean, heat, cool). Each API follows a structured signature with arguments specifying the target object and, when relevant receptacle or landmark, thereby allowing natural language instructions to be mapped to executable symbolic actions. Following the API design principles of prior works (Sarch et al., 2023; 2024), we adopt a similar modular organization to promote compositionality, extensibility, and interpretable policy synthesis.

In our evaluation scenarios, agents face a continual stream of tasks sampled in an open-ended fashion, reflecting the unpredictability of real-world environments. We evaluate FCGRAFT on a comprehensive set of 812 tasks drawn from ALFRED's instruction templates and instantiated through the API set, and we group them into two categories: simple tasks, solvable with shallow logic (e.g., *"Move a plunger to the cabinet."*), and complex tasks, which require reasoning over multiple spatial and temporal dependencies or involve state changes (e.g., *"Place a cooked apple in the refrigerator."*). This task structure allows us to evaluate FCGRAFT's ability to support scalable and reusable policy synthesis across diverse and dynamic task sequences.

*Table 7.* Instructions and executable APIs in ALFRED.

| | Template | Example |
|---|---|---|
| Instruction | Pick & Place
Stack & Place
Pick Two & Place
Clean & Place
Heat & Place
Cool & Place
Examine & in Light | *"Move a plunger to the cabinet."*
*"Drop a frying pan with a knife in it into the sink."*
*"Put the two CDs on the desk closest to the window in the safe."*
*"Place a washed pan on the counter."*
*"Place a cooked apple in the refrigerator."*
*"Place a cold tomato slice on the counter."*
*"Pick up the cell phone and look at it by the light of the lamp."* |
| API | OpenObject [Object] [Receptacle Object] | `target_lettuce = InteractionObject('Lettuce', landmark = 'Fridge')`
`target_fridge.open()` |
| | CloseObject [Object] | `target_microwave = InteractionObject('Microwave')`
`target_microwave.close()` |
| | ToggleObject [Object] | `target_lamp = InteractionObject('Lamp', landmark = 'Corner')`
`target_lamp.toggle_on()` |
| | SliceObject [Object] [Receptacle Object] | `target_apple = InteractionObject(Apple, landmark = 'CounterTop')`
`target_apple.slice()` |
| | GotoLocation [Object] [Receptacle Object] | `target_lettuce = InteractionObject('Lettuce', landmark = 'CounterTop')`
`target_lettuce.go_to()` |
| | PickupObject [Object] [Receptacle Object] | `target_lettuce = InteractionObject('Lettuce', landmark = 'CounterTop')`
`target_lettuce.pickup()` |
| | PutObject [Object] [Receptacle Object] | `target_lettuce = InteractionObject('Lettuce', landmark = 'CounterTop')`
`target_countertop = InteractionObject('CounterTop')`
`target_lettuce.place(target_countertop)` |
| | CoolObject [Object] [Receptacle Object] | `target_lettuce = InteractionObject('Lettuce')`
`target_lettuce.cool()` |
| | HeatObject [Object] [Receptacle Object] | `target_sink = InteractionObject('Sink')`
`target_sink.empty()` |
| | CleanObject [Object] [Receptacle Object] | `target_potato = InteractionObject('Potato', attributes = ['cooked'])`
`target_potato.cook()` |

## A.3. TEACh

TEACh (Padmakumar et al., 2022) spans 109 unique scenes across all 30 kitchens and most of the 30 living rooms, bedrooms, and bathrooms in AI2-THOR, comprising 3215 successful human-human gameplay sessions with rich conversational data (~45k utterances averaging 13.67 per session) and long action trajectories (averaging 131.8 Follower actions per session). The dataset covers 12 task families (e.g., put all X on Y, make coffee, prepare breakfast) with 438 parameterized variants defined through a hierarchical Task Definition Language that specifies object state changes (e.g., sliced, toasted, clean) and spatial relations. From these components, TEACh defines three evaluation settings: Execution from Dialogue History (EDH) with 11,176 instances, Trajectory from Dialogue (TfD), and Two-Agent Task Completion (TATC), with seen/unseen splits to assess generalization across novel rooms. This diversity, combined with natural dialogue supervision featuring varied instruction granularity and real-time clarification, makes TEACh particularly suitable for studying language grounding, hierarchical reasoning, long-horizon control with dialogue-based correction, and generalization across interactive household contexts.

**Environment.** For our TEACh evaluation, we use three progressively challenging scenarios that test dialogue-grounded task execution in realistic settings. (1) In *open-composition*, we examine how agents handle increasing dialogue complexity across TEACh's task hierarchy. Simple tasks like *"Water the plant."* involve minimal back-and-forth: the Commander states the goal, and the Follower executes. By contrast, compositional tasks such as *"Prepare breakfast."* require more coordination. This scenario tests whether agents can maintain coherent dialogue over extended, multi-phase interactions. (2) In *open-perturbation*, we evaluate robustness under TEACh's natural human communication patterns. The dataset contains instructional errors (e.g., *"The mug is on the table."* when it is elsewhere), temporal misalignment (utterances arriving out of order), and mid-task corrections (e.g., *"Actually, use the other sink."*). Agents must detect conflicts between instructions and observations, initiate clarification dialogue, and recover from miscommunication, reflecting deployment conditions where instructions are imperfect. (3) In *open-evolution*, we test replanning capabilities under dynamic dialogue. The dataset includes cases where the Commander revises instructions mid-execution, for example changing the goal from *"Put each tissue box on a different table."* to *"Place all the tissue boxes on the same table."*. Agents must interpret such corrections, abandon partially executed plans, and formulate new plans through continued dialogue.

TEACh's action space supports these scenarios through a dual-channel design: physical actions (navigation, manipulation, state changes) executed by the Follower, and free-form text dialogue between the agents. The Commander's special actions (`ProgressCheck`, `SearchObject`) enable task monitoring and environmental queries that support collaboration. This

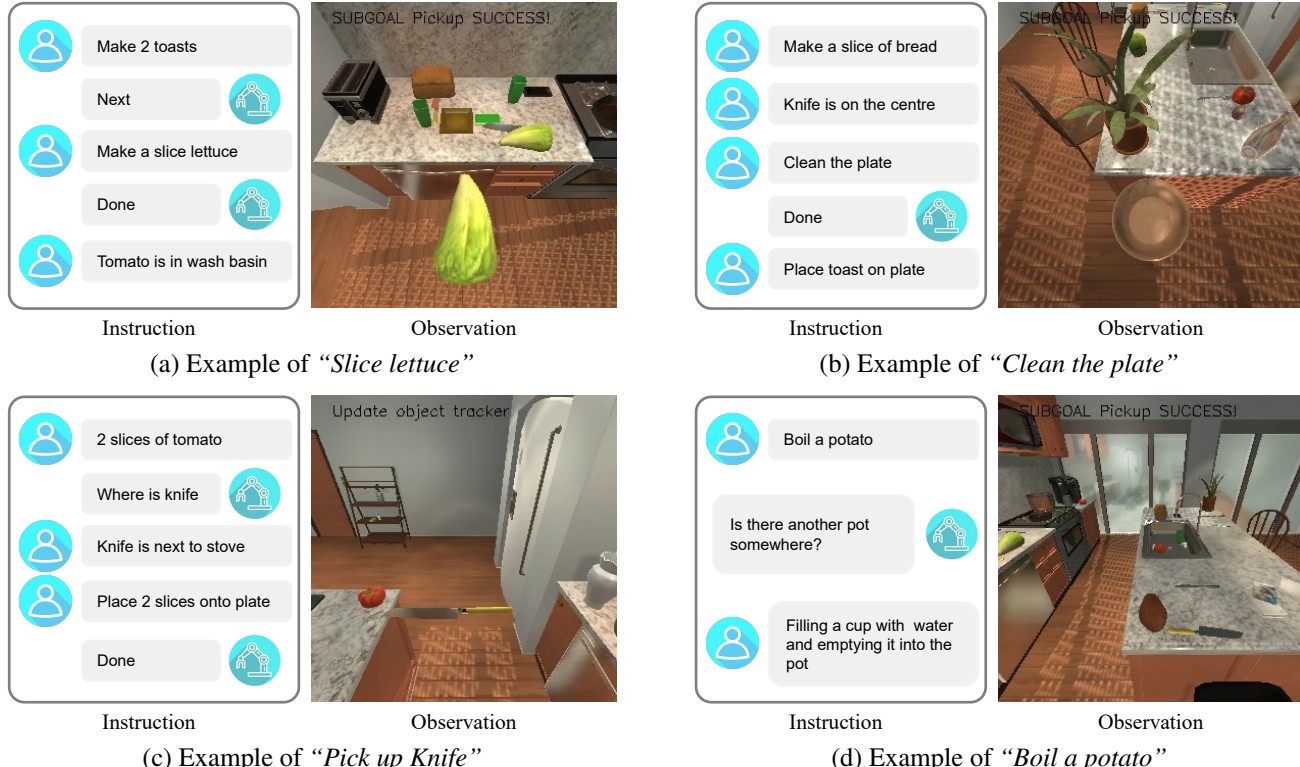

Figure 6. Examples of four task types: slice, clean, pick up, and boil.

asymmetric information structure, with task knowledge and execution distributed across agents, makes dialogue-based coordination essential in all three scenarios.

**Task.** TEACH tasks are grounded in a library of action primitives (APIs) that enable interaction with objects and receptacles in the scene. As summarized in Table 8, these APIs cover both basic object manipulation (e.g., pickup, place, open, close) and state-altering operations (e.g., clean, heat, slice, toggle). Each API follows a structured signature with arguments specifying the target object and, when relevant, the receptacle or landmark, thereby allowing natural language dialogues and instructions to be deterministically mapped to executable symbolic actions. Following the API design principles of prior works (Sarch et al., 2023; 2024), we adopt a similar modular organization to promote compositionality, extensibility, and interpretable policy synthesis.

In our evaluation scenarios, agents face a continual stream of tasks sampled in an open-ended fashion, reflecting the unpredictability of real-world collaborative environments. We evaluate FCGRAFT on 621 tasks drawn from TEACH's dialogue-based instruction templates and instantiated through the API set, and we group them into two categories: simple tasks, solvable with shallow logic and minimal dialogue context (e.g., *"Put the tomato on the counter."*), and complex tasks, which require reasoning over extended dialogue histories, handling clarification requests, and managing multi-step dependencies with state changes (e.g., *"Make breakfast - toast the bread and serve it with cleaned lettuce on a plate."*). This task structure allows us to evaluate FCGRAFT's ability to support scalable and reusable policy synthesis across diverse dialogue-driven interactions and dynamic task sequences.

*Table 8.* Instructions and executable plans in TEACh.

| | Type | Example |
|---|---|---|
| Instructions | Put all X on Y | \<Commander\>*"Put all tissue box on any side table."* |
| | Put all X in Y | \<Commander\>*"Hi. Today we are putting remote controls in a box."*
*"The box is on the couch."*
*"There is a remote control on the cabinet."* |
| | Water plant | \<Commander\>*"Water the plant."* |
| | Clean all X | \<Driver\>*"Hi, what do you need me to do?"*
\<Commander\>*"Please clean all the plates."* |
| | Make X | \<Commander\>*"Please make a salad"*
\<Driver\>*"Where do I find the thing I need?"*
\<Commander\>*"the lettuce should be in the black bin"* |
| | Prepare X in Y | \<Commander\>*"We need to prepare coffee in a clean mug."*
\<Driver\>*"Where is the mug please."*
\<Commander\>*"In the sink."* |
| | Boil X on Y | \<Commander\>*"Boil a potato."*
\<Driver\>*"Where is it?"*
\<Commander\>*"On the chair."* |
| | Cook N slices of X | \<Commander\>*"Cook 5 slices of potato and serve on a plate."* |
| | Serve N slices of X | \<Commander\>*"Make a 1 slice tomato"* |
| Plans | Pickup & Place | `target_newspaper = InteractionObject('Newspaper')`
`target_furniture = InteractionObject('Furniture')`
`target_newspaper.pickup()`
`target_newspaper.place(target_furniture)` |
| | Slice | `target_tomato = InteractionObject('Tomato', landmark = 'Sink')`
`target_tomato.go_to()`
`target_tomato.pickup()`
`target_tomato.slice()` |
| | Cook | `target_potato.pickup()`
`target_potato.go_to('Stove')`
`target_potato.boil()` |
| | Clean | `target_plate = InteractionObject('Plate', landmark = 'Sink')`
`target_plate.pickup()`
`target_plate.rinse()` |

### A.4. Real-world Test

**Environment.** Our real-world setup consisted of a 7-DoF Franka Emika Research 3 robotic arm mounted on a tabletop workspace. For perception, we deployed two Intel RealSense D435 RGB-D cameras positioned on opposite sides of the table. The dual-camera configuration provided complementary viewpoints, and their depth streams were fused into a unified point cloud to reconstruct a high-resolution 3D map of the workspace. This spatial map served as the basis for accurate object localization and enabled reliable grounding of visual observations. The perception and control modules were integrated in real time, ensuring consistent operation during manipulation tasks.

**Object detection.** Task-relevant objects were placed on the tabletop within the robot's reachable workspace. To identify and localize them, RGB images captured by both cameras were processed using the Grounding DINO model, which generated category-aware bounding boxes. These 2D detections were then projected onto the fused 3D map, allowing us to recover precise object coordinates in the robot's reference frame. The combination of semantic cues (object category) and geometric grounding (3D position) enabled robust object detection and tracking across diverse physical configurations, which was crucial for executing manipulation skills.

**Task.** Real-world tasks are executed using a set of predefined primitive skill APIs. As summarized in Table 9, a total of 10 primitive skills are defined, including basic manipulation operations such as pick, place, move, and push. Each skill is associated with task-specific parameters; for example, pick allows adjustment of the gripper force, while pull controls the distance parameter. These continuous parameters enable fine-grained control in scenarios such as grasping cups of varying shapes or pulling a drawer by a specified distance. Thus, the same primitive skill API can be flexibly adapted depending

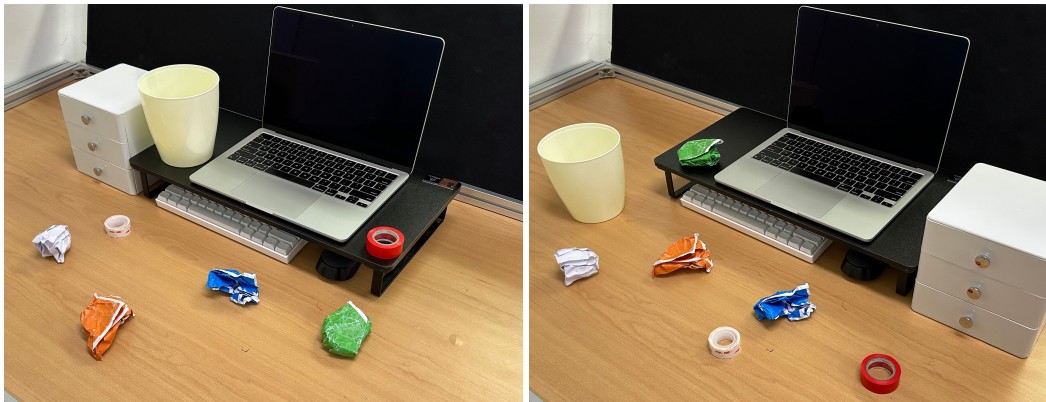

(a) Example of *Office Desk Rearrangement* environment

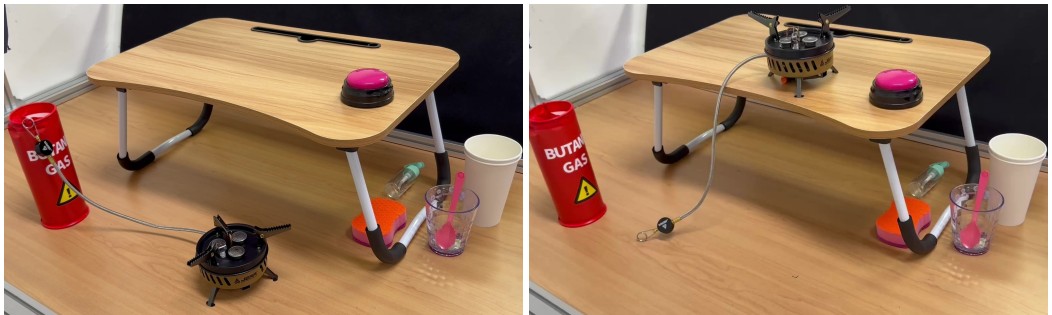

(b) Example of *Cooking Workstation Preparation* environment

*Figure 7.* Environment examples set of Real-world.

on the environment and task objectives. Real-world experiments were conducted in two environments, each consisting of multiple tasks with varying object configurations. For both environments, objects were sampled from a global object pool, and a random subset (typically 3-5 objects) was placed in randomized positions at the beginning of each trial. This randomization was performed across multiple trials (N=3 per task) to evaluate the robustness of our approach under diverse spatial configurations and object combinations.

The first environment is an office desk setup, where objects such as drawers, stationery, cups, and trash were placed on the desk. The tasks in this environment focus on organizing the workspace, including examples such as throwing paper into the trash bin or sorting stationery into a drawer. This environment was chosen to evaluate the compositionality of primitive skills and the reusability of cached code through repetitive organizing tasks within a constrained space. In particular, by repeatedly presenting tasks that are structurally similar but slightly varied, this setup assesses how FCGRAFT can efficiently reuse skills and rapidly synthesize code policy.

The second environment is a cooking workstation preparation setup, which contains diverse objects such as a gas can, burner, and cooking utensils. The tasks in this environment focus on preparatory actions for cooking, such as removing objects placed on the burner to make it usable or organizing utensils to clear the workspace. This environment was chosen to evaluate whether the agent can detect new state changes or additional sub-tasks during execution and quickly update its policy to complete the task. Unlike the office desk setup, the tasks here cannot be solved by simply reusing cached code; instead, they are designed to require cache-patching to be executed. Consequently, this environment allows us to verify how effectively and efficiently the cache-based cache-patching mechanism of FCGRAFT operates in real-world scenarios. Example images of both environments are shown in Figure 7.

*Table 9.* Instructions and executable APIs in Real-world.

| | Template | Example |
|---|---|---|
| Instruction | Move | *"Move to the button."* |
| | Press | *"Press the emergency button."* |
| | Pick & Place | *"Throw the trash into the bin."* |
| | Pick & Pull | *"Open the drawer."* |
| | Pick & Push | *"Close the drawer."* |
| | Pick & Move | *"Take the tape and move it onto the drawer."* |
| API | Ready Pose | `go_to_ready_pose()` |
| | Set Target Pose [x,y,z,r,p,y] | `setTargetPose(...)` |
| | Move to Target | `execute_go()` |
| | Pick [gripper_force, axis] | `execute_pick(gripper_force=7, axis=2)` |
| | Place [axis] | `execute_place(axis=2)` |
| | Push [gripper_force, axis, distance] | `execute_push(gripper_force=5, axis=0, distance=0.08)` |
| | Pull [gripper_force, axis, distance] | `execute_pull(gripper_force=7, axis=1, distance=0.03)` |
| | Sweep [axis, distance] | `execute_sweep(axis=0, distance=0.04)` |
| | Rotate [gripper_force, axis] | `execute_rotate(gripper_force=6, axis=2)` |
| | Gripper [width, force] | `execute_gripper(0.005, 5)` |

### A.5. Baseline

For comprehensive comparison, nine baseline methods are categorized into three groups:

- General CodeLLM-based programming methods leverage a CodeLLM to directly generate executable code policy from natural language instructions for solving embodied tasks.
  - CaP (Liang et al., 2023) introduces the Code-as-Policies paradigm, translating natural language instructions into executable robot control code.
  - SCoT (Li et al., 2025a) introduces structured chain-of-thought (CoT) prompting for reliable code generation, which we adapt to guide CodeLLM in producing code policy for embodied tasks.

- Memory-based approaches retrieve previously generated code or related textual information and supply it as additional context for policy synthesis.
  - HELPER (Sarch et al., 2023) equips embodied agents with an external memory of language–program pairs, enabling retrieval-augmented prompting for LLMs to parse open-domain instructions.
  - LRLL (Tziafas & Kasaei, 2024) is a lifelong learning framework that enables an LLM-based agent to dynamically grow and retrieve a skill library, allowing composable and generalizable policies for increasingly complex manipulation tasks.
  - PromptBook (Arenas et al., 2024) integrates examples, APIs, documentation, and CoT prompting to enable LLMs to generate code policy and acquire new low-level manipulation skills in a zero-shot manner.

- KV caching methods accelerate CodeLLM inference by reusing previously computed attention states throughout code policy generation.
  - CAG (Chan et al., 2025) proposes cache-augmented generation as an alternative to RAG by preloading relevant knowledge into the LLM's context and caching runtime parameters, thereby eliminating retrieval latency and errors while reducing system complexity.
  - RAGCache (Jin et al., 2024) introduces a multilevel dynamic caching system for RAG that stores and reuses intermediate knowledge states, reducing latency and computation costs while improving inference throughput.
  - PromptCache (Gim et al., 2024) accelerates LLM inference by caching and reusing attention states of recurring prompt segments, thereby achieving substantial latency reductions without modifying model parameters.
  - EPIC (Hu et al., 2024) introduces position-independent caching with the LegoLink algorithm to enable modular KV reuse beyond prefix matching, significantly improving LLM serving efficiency without sacrificing accuracy.

Unless otherwise noted, all baselines use the same CodeLLM configuration (max new tokens = 2048, temperature = 0.0, i.e.,

greedy decoding). When memory or cache modules are involved, we adopt the all-MiniLM-L6-v2 embedding model with cosine similarity for semantic retrieval.

**CaP.**     We adopt CaP as a baseline framework by directly prompting a CodeLLM to generate executable code policy from natural language task instructions. In our implementation, the generated code is executed in the embodied environment without additional post-processing or fine-tuning, and each trial begins from a novel prompt without access to external memory or caching. As a result, long-context prompting and high token costs can become bottlenecks in long-horizon tasks, making this configuration representative of the raw performance of CaP in our evaluation. We refer to the publicly available implementation[1].

**SCoT.**     We implement SCoT based on the code released by the authors (though the repository is no longer accessible), following their prompt design. The method uses a two-stage prompting format: first generating pseudo-code from task instructions, then producing executable code conditioned on it. This structured decomposition improves reliability compared to CaP. SCoT still follows CaP's design. It relies on long prompts for each new task without memory or reuse and is inefficient in long-horizon settings.

**HELPER.**     We adopt HELPER as released, with no changes to hyperparameters or architectural components. The system retrieves the top-3 most relevant memory entries based on cosine similarity to the current task instruction. This retrieval-augmented prompting improves adaptability to diverse instructions. HELPER does not accumulate new experiences into memory and remains sensitive to noisy or irrelevant entries, relying on long-context code generation without efficient reuse. We refer to the publicly available implementation[2].

**LRLL.**     We adopt LRLL as a baseline by implementing its core mechanisms for growing and retrieving a skill library with a CodeLLM for policy generation. Relevant skills are identified through semantic retrieval using cosine similarity (top-3) from the skill library based on the current task instruction. These skills are incorporated into the prompt context along with the task specification. The CodeLLM generates the final policy by composing the retrieved components. This design enables continual accumulation and reuse of skills through the library and provides more flexibility than approaches that rely only on transient retrieval. In practice, performance depends on how reliably the retrieved skills are composed. The experience memory also becomes more difficult to manage as it grows.

**PromptBook.**     We implement PromptBook as a baseline by constructing prompts that combine API documentation, three to five usage examples per API, and chain-of-thought templates for task decomposition, with example retrieval based on semantic similarity. The assembled prompt guides the CodeLLM through reasoning steps before generating the final code policy. Prompts are typically on the order of about two thousand tokens depending on the retrieved examples. The design scales less efficiently because task-relevant information is provided within the prompt for each new task rather than being accumulated in a reusable library.

**CAG.**     We adapt CAG, originally proposed for text-based QA, by integrating it with CaP so that the method produces executable code policy in CaP format for embodied control tasks. Successful code policy and associated skills are stored in the cache as preloaded context and reused for subsequent tasks instead of retrieving external text. The design reduces retrieval latency and errors through cached knowledge. The cache is fixed once populated and does not support dynamic updates, so adaptability to new tasks and environments is limited. We refer to the publicly available implementation[3].

**RAGCache.**     We adapt RAGCache, originally proposed as a caching framework for RAG, by implementing it in the CaP setting so that it produces executable code policy for embodied control tasks. In our implementation, generated policies are stored in the cache together with their task descriptions. The cache is incrementally updated through prefix-based indexing to support retrieval of partial matches. On a cache hit, the retrieved policy is adapted to the current task using the CodeLLM. On a miss, a new policy is generated and added to the cache for future reuse. The design enables reuse of accumulated policies over time. Prefix-based retrieval can lead to low hit rates when the cache is still small.

**PromptCache.**     We adapt PromptCache, originally introduced to accelerate inference on general text tasks, by reimplementing it in a CaP setting so that it produces executable code policy for embodied control. Our implementation caches the key–value states of recurring prompt segments, indexed by task prefixes, and reuses them for new tasks when sufficient similarity is detected. The design provides efficiency gains when prompt structures repeat. The cache is static and does

---

[1]https://github.com/google-research/google-research/tree/master/code_as_policies
[2]https://github.com/Gabesarch/HELPER
[3]https://github.com/hhhuang/CAG

not support dynamic updates, since it follows the original design around fixed prompt templates. Adaptability in diverse embodied tasks is limited. We refer to the publicly available implementation[4].

**EPIC.** We implement EPIC for embodied control by adapting its caching methodology to the CaP setting so that the system produces executable code policy. As the original implementation is not publicly available, we reproduced its approach based on the paper description. We added a preprocessing layer that transforms and aligns cached key–value states for task-specific adaptation before reuse. Cached representations are then fed to the CodeLLM to guide policy generation. The design enables flexible reuse of cached prompt information across tasks. The cache is static and does not support dynamic updates, which reduces adaptability in diverse embodied scenarios.

### A.6. Metric

Performance evaluation is conducted using diverse metrics across four complementary dimensions:

- **Task accuracy** evaluates how reliably an agent achieves task goals.
  - Task success rate (SR) measures the proportion of tasks in which all required sub-goals are successfully completed, indicating overall task-level performance (Shridhar et al., 2020).
  - Goal-conditioned success rate (GC) computes the fraction of individual sub-goals achieved across all tasks, capturing the agent's partial progress even when full task completion is not attained (Shridhar et al., 2020).
- **Computational efficiency** quantifies the inference cost of policy synthesis.
  - Policy synthesis latency (PSL) measures the average time required by a CodeLLM-based method to produce executable code policy.
  - Time to First Token (TTFT) measures the initial decoding delay before the first output token is generated.
  - Number of generated tokens (NGT) counts the total tokens generated during code policy generation.
- **Behavioral consistency** assesses coherence and transferability across tasks.
  - Code similarity (CSIM) quantifies the structural overlap between code policy generated for logically equivalent tasks (Chon et al., 2024).
  - Forward transfer (FWT) measures the improvement in performance on future tasks resulting from updating the code cache with previously encountered tasks, thereby capturing positive knowledge transfer (Lopez-Paz & Ranzato, 2017).
  - Backward transfer (BWT) measures the effect of updating the code cache with subsequent tasks on performance for previously seen tasks, indicating whether prior knowledge is preserved or forgotten (Lopez-Paz & Ranzato, 2017).
- **Memory efficiency** quantifies the effective use of GPU memory resources during policy synthesis.
  - Hit rate (HR) computes the proportion of successful cache reuses during policy synthesis.
  - GPU memory utilization (MU) quantifies the proportion of peak allocated GPU memory with respect to the available capacity (Liu et al., 2025).
- Rank is computed as a weighted sum of SR and the inverse of min–max normalized PSL, yielding a single scalar indicator that balances effectiveness and efficiency.

Table 1 and Table 2 in Section 5.2 report the Rank metric, which combines task effectiveness and computational efficiency into a single scalar score. Specifically, we first compute the average PSL for each method and apply min-max normalization across all baselines. Since lower PSL values indicate better efficiency, we use the inverse of the normalized PSL to align its direction with SR. The final Rank is then calculated as a weighted sum of SR and the inverse-normalized PSL with $\alpha = \beta = 0.5$, as defined in Eq. (7).

$$\text{Rank} = \alpha \cdot \text{SR} + \beta \cdot \left( 1 - \frac{\text{PSL} - \text{PSL}_{\min}}{\text{PSL}_{\max} - \text{PSL}_{\min}} \right) \tag{7}$$

## B. FCGRAFT: Functional Cache Grafting

### B.1. Implementation

All experiments are conducted under a fixed framework-server configuration to ensure consistent measurements of policy synthesis latency, cache management, and memory utilization. Table 10 summarizes the software stack, GPU, and random

---

[4] https://github.com/MachineLearningSystem/24MLSYS-prompt-cache

seeds used in our implementation.

*Table 10.* System configuration for the main framework server.

| Component | Specification |
| --- | --- |
| Python | 3.9.18 |
| CUDA / cuDNN | CUDA 12.4 / cuDNN 9.1 |
| PyTorch | 2.5.1 |
| Transformers | 4.46.3 |
| GPU | NVIDIA RTX 4090 (24GB) |
| Random Seed | 3 runs with seeds 1, 2, 3 |

**System architecture.** The framework server coordinates prompt orchestration, code generation, cache access, and feedback-guided regeneration. The simulation server executes the generated code policy within the environment and returns structured feedback to the framework server. The two servers communicate asynchronously via REST APIs with JSON message passing. Control is framework-driven in RLBench and simulation-initiated in ALFRED and TEACh, reflecting differences in the benchmark interfaces. Algorithm 1 summarizes the control flow.

**Module-level cache reuse.** FCGRAFT constructs the cache library at the granularity of reusable functions rather than arbitrary token spans. Each function cache entry is split into a Function-Interface tier and a Function-Code tier. The Function-Interface tier stores the textual interface and its KV state, which are used during cache-stitching to provide compact function-level context for policy synthesis. The Function-Code tier stores the full implementation and its KV state, which are used during execution and localized repair. This separation allows the model to reuse compact interface-level modules during policy generation, while preserving validated implementation-level code for execution.

This module-level design is important for embodied policy synthesis because robotic tasks often reuse the same primitive skills under different object, scene, and parameter configurations. Rather than regenerating or re-encoding every function description for each new instruction, FCGRAFT composes the relevant cached interfaces and links the corresponding validated implementations. Therefore, cached KV states serve not only as serving-time acceleration artifacts, but also as reusable policy-memory units aligned with executable skill boundaries.

**Implementation details.** The implementation details below specify how FCGRAFT is instantiated in practice:

- **Prompt construction.** Prompts include task instructions and observations under a fixed header and canonical entry point. They also contain function documentation and example implementations that can be reused during replanning. We support multiple observation modalities, including natural-language instructions in RLBench, scene descriptions in ALFRED, dialogue in TEACh, and symbolic states in the real-world setup.

> **Prompt example**
>
> ```
> # MANDATORY: Use these EXACT objects and skills. NO placeholders!
> # Available objects: 'block', 'target', 'success'
> # Available skills: move, pick, place, push, open_gripper, close_gripper
> # IMPORTANT for button/switch tasks: close_gripper first, then push/press
> # For this specific task, start with: push, 'block')
> # Then continue with the remaining implementation.
> # If a line may cause errors or need revision,
> # insert '# ERROR_FLAG' immediately before it.
>
> # Starting code (continue from here):
> obs, reward, done = run_action(skill, object, offset(if needed))
> <task instruction text continues here...>
> ```

- **Clarification of locality score components.** The locality score comprises three components-$\ell_{\text{freq}}$, $\ell_{\text{asso}}$, and $\ell_{\text{sema}}$-each

designed to capture a distinct aspect of behavioral regularities that arise in open-domain task streams. These terms are not intended to preserve recency alone; rather, they reflect temporal repetition, compositional dependencies, and semantic distinctiveness. Formal definitions are provided below.

– $\ell_{\text{freq}}$ *(Usage frequency).* This term measures the relative frequency with which a function is invoked over recent tasks:

$$\ell_{freq}(f_k) = \frac{\text{count}(f_k)}{\max_j \text{count}(f_j)} \tag{8}$$

It captures repetitive or recurrent usage patterns that arise during continuous task execution.

– $\ell_{\text{asso}}$ *(Conditional association).* This component quantifies compositional relationships by measuring the conditional likelihood that $f_j$ is invoked following $f_k$ within an execution trace:

$$\ell_{asso}(f_j \mid f_k) = \frac{\text{cooccur}(f_k, f_j)}{\text{count}(f_k)} \tag{9}$$

It reflects order-dependent transitions and functional co-occurrence patterns that frequently appear in multi-step procedures.

– $\ell_{\text{sema}}$ *(Semantic novelty).* This term evaluates the semantic distinctiveness of a function by computing the perplexity of its implementation under the CodeLLM:

$$\ell_{sema}(f_k) = \frac{\text{PPL}(f_k)}{\max_j \text{PPL}(f_j)} \tag{10}$$

It promotes the retention of semantically unique or infrequent functions, supporting robustness to atypical or unpredictable task variations.

- **Cache-stitching.** During the initial generation, we inject KV states from *Function Interface tier* ($\mathcal{I}$) entries as additional context. *Function Code tier* entries ($\mathcal{C}$) are excluded at this stage to avoid redundant reasoning over full implementations. In subsequent generations, ($\mathcal{C}$) KV states are included, but only for those functions whose ($\mathcal{I}$) entries were actually used. The model decodes greedily and stops at predefined stop phrases or EOS. Stop phrases include markers such as '# code_end' or 'def'. The complete list is provided in the released code. Generated lines are appended after the canonical entry point and must be executable code. Non-executable content such as comments or placeholders is filtered by the stop rules.

- **Execution feedback.** During policy execution, structured feedback is recorded whenever the generated policy either fails to execute or does not satisfy task requirements. Feedback spans from low-level interpreter or API errors, to skill-level execution failures, to higher-level semantic violations and unmet goals. This design ensures that cache-patching is not limited to repairing syntactic errors, but also incorporates semantic corrections such as adjusting object usage or addressing unsatisfied subgoals. Representative feedback categories are illustrated in the code release.

- **Cache-patching.** Before execution, undefined function calls or missing parameters are flagged during generation. After generation, the flagged spans are corrected first, and then the program is executed. During execution, if a runtime exception occurs, the error is localized at the line level and the program is split into prefix, error, and suffix. We apply *fill-in-the-middle* guided by a short structured CoT trace, reusing the prefix KV states and regenerating only the middle span, which is then concatenated with the suffix text. The CoT trace is discarded before re-execution, and only the corrected code is run. Cache-patching after execution is limited to three retries. The same refinement mechanism is applied across all benchmarks with adaptations for each observation type.

---

**Example of cache-patching in real-world**

**Prompt:**

```
# Task: {instruction}
# IMPORTANT: Functions from robot_tasks are already imported.
Just call them directly, DO NOT redefine them.
# For other tasks without matching functions,
```

```
use basic skills (setTargetPose, execute_pick, execute_place)
# If a line may cause errors or need revision,
# insert '# ERROR_FLAG' immediately before it.

def robot_move(self):
        self.ps.go_to_ready_pose()
        # First handle the trash task using the imported function
```

**Generated code:**

```
...
push_button_task(self.ps, self.get_obj)
...
```

**Error feedback (as provided by API)**

```
# Task instruction:
Push the gas_manage_button to stop the gas leak.

# Generated robot_move implementation:
self.ps.go_to_ready_pose()
push_button_task(self.ps, self.get_obj)
self.ps.go_to_ready_pose()

# Known discrepancies:
Missing objects: button

# Think step-by-step about why the generated code fails and how to correct it.
Respond in strict JSON with keys:
  "reasoning": ordered list of short reasoning steps,
  "fixes": ordered list of concrete code edits or high-level actions,
  "verdict": brief summary of the main failure reason.
Let's reason step by step.
```

**Result of cache-patching:**

push_button_task(self.ps, self.get_obj, + button_obj="gas_manage_button")

- **Unified exception handling.** Although the internal mechanisms differ between simulation and real-world execution, we standardize all failure signals into a unified exception interface. Interpreter-level exceptions (e.g., `SyntaxError`, `NameError`) arise from Python execution and behave identically across environments, while environment-level exceptions (`RobotError`) capture grasp failures, workspace violations, and other skill-level issues. Table 11 summarizes the interpreter-level and environment-level exception types handled by the unified interface. In simulation, these failures are reported directly through the native APIs. In real-world deployment with the Franka Emika Research 3, we detect failures using depth cameras, object detectors, and a VLM-based state check. For example, unsuccessful grasps or unmet semantic conditions (e.g., drawer not opened) trigger a `RobotError` when the observation does not match the expected postcondition. All exceptions-syntactic, execution-level, or perception-driven-are funneled through the same interface, enabling cache-patching to resolve them uniformly. Thus, the try-except block in Algorithm 1 is not simulation-specific; it reflects an environment-agnostic design that consistently supports both simulated and real-world execution.

### B.2. Hyperparameter

**LLM settings.** We use `Qwen2.5-Coder-14B` via Hugging Face with 8-bit quantization. For size ablations on the primary code model we evaluate `Qwen2.5-Coder` at {3B, 7B, 14B}. For cross-family ablations we fix the scale at 7B and compare the families Qwen2.5, Gemma, Llama 2, and DeepSeek; we additionally evaluate 7B general-purpose LLMs

*Table 11.* Interpreter-level and environment-level exceptions handled by the unified exception interface.

| Type | Category | Details |
|---|---|---|
| Interpreter-level | SyntaxError | Invalid Python syntax
Malformed code block
Non-code text inside code region |
| Interpreter-level | ClassNotFound | Missing Pygments lexer for code block
Fallback to generic text mode
Unsupported code type detected |
| Interpreter-level | FileNotFoundError | Missing object metadata file
Missing orientation mapping file
Missing noise parameter configuration |
| Environment-level | RobotError | Skill sequence execution failure
Path out of workspace
Object out of reach
Grasp or manipulation failure
Timeout during primitive execution
Invalid object type binding
Missing or undefined object handle
Missing constraints in object mapping
Infeasible approach pose
Orientation constraint violation
Safety stop or low-level controller failure |

as drop-in code generators. All decoding and framework-level hyperparameters are held fixed as in Table 12.

*Table 12.* Hyperparameters (decoding and framework-level).

| **Model generation hyperparameters** | |
|---|---|
| max_new_tokens | 2048 |
| temperature | 0.0 (greedy) |
| top-$k$, top-$p$ | N/A (greedy; not used) |
| **Framework-level hyperparameters** | |
| Eviction threshold (perplexity-based) | $\tau = 15.0$ |
| Locality weights ($\alpha$: temporal, $\beta$: spatial, $\gamma$: semantic) | $\alpha = 0.4$, $\beta = 0.3$, $\gamma = 0.3$ |
| Execution trace length | 20 |
| Temporal decay rate | 0.01 |
| GPU memory threshold (%) / reserve (GB) | 0.85 / 2.0 |
| Min free GPU before generation (GB) | 1.5 |
| Top-$N$ blocks used | 2 |
| FIM repair: max_tokens | 128 |

## C. Additional experimental result

### C.1. Benchmark experiment details

Table 13 expands on Table 1 in Section 5.2 by reporting detailed results across all metrics for each open-domain scenario in the ALFRED benchmark. In Table 13a, FCGRAFT achieves the highest SR, improving by 5.30% over LRLL. For GC, it surpasses LRLL by 3.72%. In terms of PSL, where lower is better, FCGRAFT reduces it by $2.08\times$ compared to CAG. NGT is also significantly reduced by 58.18 tokens relative to RAGCache. All cache-based baselines and FCGRAFT begin with the same initial KV cache states derived from basic success code policies. CAG, EPIC, and PromptCache do not update the cache afterward, so no eviction occurs and the hit ratio remains 100%. RAGCache and FCGRAFT keep inserting new entries after initialization, which reduces the hit ratio. Nevertheless, FCGRAFT excels in MU, achieving 10.22% higher than RAGCache. In Table 13b, FCGRAFT achieves the highest SR, improving by 2.53% over LRLL. For GC, it surpasses LRLL by 3.77%. In terms of PSL, where lower is better, FCGRAFT reduces it by $1.96\times$ compared to CAG. NGT is also significantly reduced by 34.85 tokens relative to PromptCache. Furthermore, FCGRAFT excels in MU, achieving 6.67% higher than RAGCache. In Table 13c, FCGRAFT achieves the highest SR, improving by 4.29% over LRLL. For GC, it

surpasses LRLL by $4.65\%$. In terms of PSL, where lower is better, FCGRAFT reduces it by $1.97\times$ compared to CAG. NGT increases by 38.53 tokens compared to CaP. Furthermore, FCGRAFT excels in MU, achieving $4.83\%$ higher than RAGCache.

*Table 13.* Extended results from ALFRED benchmark evaluation on open-domain embodied tasks.

*(a) Open-Composition* scenario.

| Method | SR | GC | PSL | NGT | HR | MU |
|---|---|---|---|---|---|---|
| CaP | 46.48±1.87 | 66.03±2.72 | 16.21±1.03 | 186.43±5.30 | - | 24.87±0.38 |
| SCOT | 47.93±0.87 | 68.38±2.38 | 30.02±3.38 | 302.41±8.02 | - | 26.43±0.82 |
| HELPER | 53.83±0.39 | 76.03±3.23 | 16.82±1.20 | 175.83±6.23 | - | 25.39±1.02 |
| LRLL | 56.28±3.20 | 78.32±2.03 | 16.02±2.02 | 198.41±12.04 | - | 26.20±1.27 |
| PromptBook | 53.94±3.96 | 75.81±2.30 | 31.34±1.82 | 274.40±7.32 | - | 28.43±2.23 |
| CAG | 38.24±0.04 | 55.27±0.23 | 12.12±0.62 | 174.03±3.94 | 100.00±0.00 | 47.88±0.03 |
| RAGCache | 39.29±1.70 | 58.43±1.82 | 13.48±0.93 | 162.48±2.73 | 58.93±3.04 | 75.86±0.49 |
| EPIC | 43.38±1.02 | 62.01±2.20 | 14.02±1.04 | 172.46±5.71 | 100.00±0.00 | 53.03±1.02 |
| PromptCache | 40.37±0.37 | 59.06±3.29 | 12.82±0.86 | 173.51±5.91 | 100.00±0.00 | 50.47±2.22 |
| FCGRAFT | 61.58±1.86 | 82.04±2.22 | 5.82±0.57 | 104.30±5.32 | 75.46±1.21 | 86.08±0.67 |

*(b) Open-Perturbation* scenario.

| Method | SR | GC | PSL | NGT | HR | MU |
|---|---|---|---|---|---|---|
| CaP | 42.23±0.12 | 60.83±0.32 | 16.15±0.23 | 184.53±0.42 | - | 24.23±1.02 |
| SCOT | 43.46±1.06 | 64.17±1.52 | 32.39±1.12 | 325.12±8.29 | - | 27.82±1.92 |
| HELPER | 52.49±0.24 | 73.82±0.52 | 15.94±0.69 | 189.62±3.02 | - | 26.39±1.93 |
| LRLL | 54.70±0.87 | 75.00±1.28 | 16.40±1.24 | 214.95±4.22 | - | 27.51±2.39 |
| PromptBook | 52.63±1.09 | 73.89±1.02 | 33.29±1.52 | 364.82±5.19 | - | 30.12±2.86 |
| CAG | 36.03±0.32 | 53.02±0.65 | 12.38±0.92 | 182.81±2.74 | 100.00±0.00 | 47.54±2.21 |
| RAGCache | 36.33±0.02 | 53.25±0.08 | 13.15±0.12 | 184.74±2.04 | 52.45±0.32 | 75.78±0.23 |
| EPIC | 37.08±1.02 | 56.66±1.22 | 13.74±0.69 | 178.29±5.72 | 100.00±0.00 | 54.83±0.32 |
| PromptCache | 36.92±2.22 | 55.12±1.95 | 12.93±1.48 | 163.35±8.92 | 100.00±0.00 | 51.25±1.08 |
| FCGRAFT | 57.23±0.23 | 78.77±0.66 | 6.32±0.51 | 128.50±2.20 | 71.04±1.00 | 82.45±0.81 |

*(c) Open-Evolution* scenario.

| Method | SR | GC | PSL | NGT | HR | MU |
|---|---|---|---|---|---|---|
| CaP | 37.22±1.72 | 55.29±2.29 | 16.28±1.20 | 165.81±6.03 | - | 26.12±2.22 |
| SCOT | 39.00±0.98 | 57.77±1.36 | 32.04±2.10 | 334.68±7.03 | - | 27.83±2.92 |
| HELPER | 47.75±1.08 | 68.92±1.52 | 16.42±0.92 | 214.32±4.55 | - | 27.92±0.76 |
| LRLL | 51.60±0.30 | 70.81±1.00 | 16.63±1.03 | 214.82±2.11 | - | 28.23±0.76 |
| PromptBook | 50.49±0.51 | 70.54±0.78 | 33.49±2.07 | 368.90±7.90 | - | 31.83±0.96 |
| CAG | 32.18±1.12 | 47.75±1.51 | 12.39±0.92 | 174.42±3.30 | 100.00±0.00 | 50.95±0.93 |
| RAGCache | 33.83±1.53 | 48.44±2.02 | 13.02±1.15 | 188.12±6.32 | 48.67±3.20 | 73.54±1.82 |
| EPIC | 34.23±0.23 | 51.48±1.52 | 13.49±1.14 | 177.23±5.81 | 100.00±0.00 | 51.32±1.22 |
| PromptCache | 34.14±0.82 | 50.22±1.62 | 13.18±1.15 | 173.58±4.26 | 100.00±0.00 | 47.83±1.92 |
| FCGRAFT | 55.89±0.85 | 75.46±1.71 | 6.29±1.25 | 204.34±3.21 | 70.26±1.82 | 78.37±1.21 |

Table 14 expands on Table 1 in Section 5.2 by reporting detailed results across all metrics for each open-domain scenario in the TEACh benchmark. In Table 14a, FCGRAFT achieves the highest SR, improving by 9.60% over LRLL. For GC, it surpasses LRLL by 8.00%. In terms of PSL, where lower is better, FCGRAFT reduces it by 2.72× compared to CAG. NGT is also significantly reduced by 79.60 tokens relative to HELPER. Furthermore, FCGRAFT excels in MU, achieving 10.95% higher than RAGCache. In Table 14b, FCGRAFT achieves the highest SR, improving by 9.48% over LRLL. For GC, it surpasses LRLL by 8.20%. In terms of PSL, where lower is better, FCGRAFT reduces it by 2.50× compared to CAG. NGT is also significantly reduced by 85.13 tokens relative to CaP. Furthermore, FCGRAFT excels in MU, achieving 12.14% higher than RAGCache. In Table 14c, FCGRAFT achieves the highest SR, improving by 11.08% over LRLL. For GC, it surpasses LRLL by 13.55%. In terms of PSL, where lower is better, FCGRAFT reduces it by 2.11× compared to CAG. NGT is also significantly reduced by 83.40 tokens relative to HELPER. Furthermore, FCGRAFT excels in MU, achieving 0.64% higher than RAGCache.

*Table 14.* Extended results from TEACh benchmark evaluation on open-domain embodied tasks.

*(a) Open-Composition* scenario.

| Method | SR | GC | PSL | NGT | HR | MU |
|---|---|---|---|---|---|---|
| CaP | 43.92±0.81 | 65.08±1.11 | 17.29±0.73 | 208.39±5.67 | - | 24.84±1.07 |
| SCOT | 44.03±1.47 | 65.73±1.52 | 33.00±0.91 | 316.82±8.09 | - | 27.10±2.89 |
| HELPER | 47.14±0.63 | 69.45±1.12 | 17.31±1.08 | 201.81±3.28 | - | 26.28±1.05 |
| LRLL | 48.99±1.73 | 71.20±2.15 | 17.82±0.82 | 221.39±2.10 | - | 26.71±1.22 |
| PromptBook | 47.08±2.24 | 70.85±2.86 | 36.92±1.47 | 355.11±12.92 | - | 30.29±2.71 |
| CAG | 35.03±2.01 | 60.39±2.41 | 14.23±1.28 | 232.59±4.31 | 100.00±0.00 | 47.66±3.04 |
| RAGCache | 35.23±0.05 | 57.03±0.12 | 14.97±0.15 | 252.44±1.08 | 48.48±0.32 | 77.28±0.84 |
| EPIC | 36.39±0.64 | 59.98±3.07 | 15.43±0.44 | 226.95±0.91 | 100.00±0.00 | 50.83±0.61 |
| PromptCache | 36.15±0.86 | 59.28±0.92 | 14.88±0.29 | 217.50±1.30 | 100.00±0.00 | 52.71±0.83 |
| FCGRAFT | 58.59±0.39 | 79.20±0.64 | 5.23±0.09 | 122.21±5.73 | 76.24±0.39 | 88.23±0.48 |

*(b) Open-Perturbation* scenario.

| Method | SR | GC | PSL | NGT | HR | MU |
|---|---|---|---|---|---|---|
| CaP | 42.21±0.28 | 63.83±0.63 | 17.23±0.39 | 212.24±2.75 | - | 25.09±0.38 |
| SCOT | 42.49±1.02 | 65.31±1.98 | 33.94±1.92 | 292.84±5.09 | - | 28.39±1.08 |
| HELPER | 44.05±0.29 | 67.19±0.83 | 17.39±1.02 | 224.00±0.08 | - | 26.93±0.62 |
| LRLL | 46.78±0.91 | 68.29±0.92 | 17.93±1.22 | 226.54±1.02 | - | 27.36±1.40 |
| PromptBook | 44.85±1.00 | 66.98±1.47 | 35.04±1.24 | 363.71±5.30 | - | 30.29±1.22 |
| CAG | 33.89±0.92 | 57.00±1.02 | 14.02±0.24 | 214.29±2.40 | 100.00±0.00 | 49.01±2.09 |
| RAGCache | 34.08±0.04 | 57.56±0.98 | 15.51±0.42 | 222.30±2.20 | 49.28±0.43 | 76.89±0.00 |
| EPIC | 34.39±0.51 | 61.03±0.58 | 15.66±0.48 | 212.71±1.12 | 100.00±0.00 | 51.38±0.35 |
| PromptCache | 33.12±0.12 | 59.42±0.39 | 14.94±1.31 | 224.53±2.04 | 100.00±0.00 | 45.84±0.00 |
| FCGRAFT | 56.26±1.09 | 76.49±1.32 | 5.61±1.83 | 127.11±6.21 | 74.02±1.18 | 89.03±1.12 |

*(c) Open-Evolution* scenario.

| Method | SR | GC | PSL | NGT | HR | MU |
|---|---|---|---|---|---|---|
| CaP | 38.07±0.12 | 58.97±0.33 | 17.03±1.20 | 239.60±5.20 | - | 24.01±1.08 |
| SCOT | 37.91±0.41 | 56.49±0.39 | 35.03±1.58 | 312.49±5.23 | - | 30.41±1.29 |
| HELPER | 40.00±0.38 | 61.47±1.05 | 17.48±0.71 | 211.60±3.04 | - | 32.70±0.24 |
| LRLL | 41.85±0.02 | 61.84±0.12 | 18.03±0.39 | 214.48±0.87 | - | 32.03±0.08 |
| PromptBook | 41.07±0.32 | 62.78±0.48 | 36.82±0.49 | 378.14±8.39 | - | 36.16±0.89 |
| CAG | 30.58±0.33 | 52.93±0.38 | 14.82±0.11 | 234.12±0.91 | 100.00±0.00 | 50.59±0.20 |
| RAGCache | 29.45±1.23 | 53.41±1.87 | 16.07±0.29 | 218.00±0.83 | 43.41±0.33 | 81.18±0.64 |
| EPIC | 33.92±1.52 | 58.75±1.74 | 17.18±0.54 | 226.10±2.22 | 100.00±0.00 | 52.83±1.04 |
| PromptCache | 31.07±1.49 | 55.02±0.49 | 16.32±0.24 | 217.6±0.92 | 100.00±0.00 | 51.71±0.39 |
| FCGRAFT | 52.93±0.62 | 75.39±0.87 | 7.03±0.82 | 128.20±5.09 | 70.92±1.02 | 81.82±0.39 |

Table 15 expands on Table 1 in Section 5.2 by reporting detailed results across all metrics for each open-domain scenario in the RLBench benchmark. In Table 15a, FCGRAFT achieves the highest SR, improving by 9.80% over LRLL. In terms of PSL, where lower is better, FCGRAFT reduces it by 2.63× compared to CAG. NGT is also significantly reduced by 14.38 tokens relative to LRLL. Furthermore, FCGRAFT excels in MU, achieving 8.29% higher than RAGCache. In Table 15b, FCGRAFT achieves the highest SR, improving by 9.00% over LRLL. In terms of PSL, where lower is better, FCGRAFT reduces it by 2.42× compared to CAG. NGT is also significantly reduced by 16.81 tokens relative to LRLL. Furthermore, FCGRAFT excels in MU, achieving 3.41% higher than RAGCache. In Table 15c, FCGRAFT achieves the highest SR, improving by 1.53% over LRLL. In terms of PSL, where lower is better, FCGRAFT reduces it by 1.84× compared to CAG. NGT is also significantly reduced by 51.60 tokens relative to CaP. Furthermore, FCGRAFT reports slightly lower MU compared to RAGCache.

*Table 15.* Extended results from RLBench benchmark evaluation on open-domain embodied tasks.

*(a) Open-Composition scenario.*

| Method | SR | PSL | NGT | HR | MU |
|---|---|---|---|---|---|
| CaP | 31.23±3.32 | 11.39±0.53 | 123.43±3.55 | - | 23.08±0.88 |
| SCOT | 32.32±2.22 | 22.03±2.03 | 246.32±12.43 | - | 27.48±2.33 |
| HELPER | 35.42±1.32 | 11.48±0.23 | 154.18±1.45 | - | 26.48±1.93 |
| LRLL | 36.11±0.00 | 10.83±0.82 | 120.00±2.15 | - | 26.43±0.00 |
| PromptBook | 35.93±1.12 | 23.49±0.63 | 206.48±3.23 | - | 28.23±0.73 |
| CAG | 26.37±2.03 | 8.53±0.83 | 157.52±10.30 | 100.00±0.00 | 44.95±4.50 |
| RAGCache | 27.26±1.08 | 9.09±0.63 | 149.35±5.02 | 59.02±1.03 | 75.34±4.83 |
| EPIC | 29.24±2.88 | 9.48±1.20 | 163.81±5.32 | 100.00±0.00 | 48.39±2.22 |
| PromptCache | 28.33±1.73 | 8.99±0.39 | 272.90±1.67 | 100.00±0.00 | 47.33±1.01 |
| FCGRAFT | 45.91±4.37 | 3.24±0.65 | 105.62±3.09 | 73.65±2.15 | 83.63±3.12 |

*(b) Open-Perturbation scenario.*

| Method | SR | PSL | NGT | HR | MU |
|---|---|---|---|---|---|
| CaP | 30.46±0.02 | 11.23±0.93 | 132.84±10.82 | - | 23.28±2.83 |
| SCOT | 30.32±0.28 | 21.37±1.84 | 189.23±15.64 | - | 29.32±3.85 |
| HELPER | 34.84±0.03 | 11.38±0.06 | 159.08±1.45 | - | 28.34±1.75 |
| LRLL | 33.82±0.02 | 12.01±0.11 | 122.81±0.78 | - | 26.47±2.30 |
| PromptBook | 33.03±0.67 | 24.03±3.89 | 192.93±12.68 | - | 28.32±0.93 |
| CAG | 24.89±1.21 | 8.39±0.37 | 221.47±3.84 | 100.00±0.00 | 45.49±0.82 |
| RAGCache | 27.03±1.04 | 9.28±0.66 | 238.79±5.38 | 57.38±1.08 | 78.93±2.02 |
| EPIC | 29.04±0.83 | 9.48±0.32 | 208.45±3.04 | 100.00±0.00 | 48.29±2.23 |
| PromptCache | 27.41±0.81 | 8.97±0.35 | 250.40±5.83 | 100.00±0.00 | 47.55±1.42 |
| FCGRAFT | 42.82±2.64 | 3.47±0.16 | 106.00±5.65 | 70.73±2.73 | 82.34±1.04 |

*(c) Open-Evolution scenario.*

| Method | SR | PSL | NGT | HR | MU |
|---|---|---|---|---|---|
| CaP | 26.84±1.20 | 11.23±0.24 | 152.62±5.03 | - | 24.62±1.03 |
| SCOT | 27.11±0.37 | 22.83±1.42 | 189.36±5.23 | - | 28.54±1.40 |
| HELPER | 31.12±1.08 | 12.16±1.42 | 174.00±3.33 | - | 28.81±0.02 |
| LRLL | 32.45±0.94 | 12.17±2.01 | 198.83±0.53 | - | 28.58±2.32 |
| PromptBook | 31.65±3.32 | 25.93±2.09 | 231.96±8.39 | - | 31.34±3.83 |
| CAG | 22.82±0.73 | 8.08±1.91 | 178.81±5.39 | 100.00±0.00 | 44.39±2.93 |
| RAGCache | 23.79±1.20 | 9.02±0.18 | 183.11±4.23 | 53.49±1.27 | 82.79±0.34 |
| EPIC | 25.85±1.25 | 9.28±0.47 | 182.31±6.34 | 100.00±0.00 | 48.81±0.53 |
| PromptCache | 25.03±0.85 | 9.12±0.65 | 193.77±3.78 | 100.00±0.00 | 47.93±0.75 |
| FCGRAFT | 33.98±1.27 | 4.39±0.43 | 101.02±3.02 | 70.29±3.21 | 82.27±6.27 |

## C.2. Real-world experiment details

The real-world evaluation is conducted over two open-domain manipulation scenarios: *Office Desk Rearrangement* and *Cooking Workstation Preparation*. For each scenario, we construct 9 distinct task conditions rather than repeating a single fixed setup. These conditions vary task composition, object identities, object placements, and workspace configurations. Each condition is evaluated over 3 randomized runs, resulting in 27 physical robot trials per scenario. This protocol is designed to evaluate whether each method remains robust under realistic variation in task layout and execution conditions, while keeping the overall number of physical trials feasible. Table 16 expands on Table 2 in Section 5.2 by reporting detailed results across all metrics for each open-domain scenario in real-world deployment. Table 17 further reports Wilson score confidence intervals for SR to clarify the reliability of the real-world evaluation.

In Table 16a, FCGRAFT achieves the highest SR, improving by 22.22% over LRLL. In terms of PSL, where lower is better, FCGRAFT reduces it by 2.39× compared to RAGCache. NGT is also significantly reduced by 53.66 tokens relative to RAGCache. Furthermore, FCGRAFT excels in MU, achieving 10.26% higher than RAGCache. In Table 16b, FCGRAFT achieves the highest SR, improving by 25.92% over LRLL. In terms of PSL, where lower is better, FCGRAFT reduces it by 3.38× compared to RAGCache. NGT is also significantly reduced by 47.22 tokens relative to RAGCache. Furthermore, FCGRAFT excels in MU, achieving 7.90% higher than RAGCache.

*Table 16.* Extended results from Real-world robotic manipulation.

*(a) Office Desk Rearrangement.*

| Method | SR | GC | PSL | NGT | HR | MU |
|---|---|---|---|---|---|---|
| CaP | 33.33±0.00 | 44.44±0.00 | 11.94±1.17 | 148.02±3.30 | - | 25.91±1.93 |
| LRLL | 55.56±11.11 | 60.74±9.25 | 12.31±0.74 | 146.39±5.83 | - | 29.04±0.84 |
| RAGCache | 44.44±22.22 | 53.09±19.21 | 9.08±0.74 | 113.33±3.86 | 87.45±2.39 | 77.04±3.02 |
| FCGRAFT | 77.78±11.11 | 81.48±13.58 | 3.80±0.31 | 59.67±5.25 | 92.60±3.23 | 87.30±3.20 |

*(b) Cooking Workstation Preparation.*

| Method | SR | GC | PSL | NGT | HR | MU |
|---|---|---|---|---|---|---|
| CaP | 51.85±6.42 | 54.32±2.14 | 13.07±1.09 | 154.02±5.30 | - | 26.31±1.83 |
| LRLL | 55.56±0.00 | 55.56±0.00 | 12.65±1.14 | 148.05±2.04 | - | 29.82±1.28 |
| RAGCache | 37.04±6.42 | 37.04±6.42 | 9.63±1.01 | 120.49±6.39 | 62.05±1.03 | 67.93±3.72 |
| FCGRAFT | 81.48±12.83 | 82.81±10.52 | 2.85±0.54 | 73.27±3.29 | 72.72±3.02 | 75.83±3.20 |

We further report Wilson score confidence intervals for the real-world success rate to clarify the reliability of the real-world evaluation under a modest number of physical trials. Each real-world scenario consists of 9 distinct task conditions, and each condition is evaluated over 3 randomized runs, resulting in 27 trials per scenario. The confidence intervals show that FCGRAFT maintains a substantial performance advantage in both real-world scenarios, although the limited number of physical trials remains a practical constraint of the real-world evaluation.

*Table 17.* Analysis on real-world success reliability across physical trials.

| Method | Office Desk Rearrangement | | Cooking Workstation Preparation | |
|---|---|---|---|---|
| | SR | 95% CI | SR | 95% CI |
| CaP | 33.33±0.00 | [18.6, 52.2] | 51.85±6.42 | [34.0, 69.3] |
| LRLL | 55.56±11.11 | [37.3, 72.4] | 55.56±0.00 | [37.3, 72.4] |
| RAGCache | 44.44±22.22 | [27.6, 62.7] | 37.04±6.42 | [21.5, 55.8] |
| FCGRAFT | 77.78±11.11 | [59.2, 89.4] | 81.48±12.83 | [63.3, 91.8] |

### C.2.1. OFFICE DESK REARRANGEMENT

The first real-world environment is an office desk setup designed around a long-horizon task of cleaning and organizing the workspace. This section presents a detailed description of the environment illustrated in Figure 3. The task is decomposed

into three sequential subtasks. In the first subtask, the agent picks up two trash items and places them into the trash bin. The second subtask requires sorting stationery into the top drawer, demonstrating the agent's ability to handle container interactions. Finally, in the third subtask, the agent disposes of the remaining trash into the bin and organizes the leftover stationery into the middle drawer. This sequence evaluates the compositionality of primitive skills such as pick, place, and pull, while also assessing how cached code can be efficiently reused across structurally similar operations. By combining repetitive but slightly varied object interactions, this environment highlights FCGRAFT's capability to synthesize consistent code policy for long-horizon desk-cleaning tasks. Figure 8 provides images of the experiment setup and execution sequence.

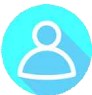
Put trash in the bin and office supplies in the drawer.

**Sub-task 1**

**Sub-task 2**

**Sub-task 3**

*Figure 8.* Real-world *Office Desk Rearrangement*. The full task sequence is decomposed into three subtasks: (1) picking up two trash items and throwing them into the bin, (2) organizing stationery into the top drawer, and (3) disposing of remaining trash and placing the leftover stationery into the middle drawer.

### C.2.2. COOKING WORKSTATION PREPARATION

The second real-world environment is a cooking workstation setup. This section provides the detailed description of the environment shown in Figure 1. The task is decomposed into three subtasks. In the first subtask, the agent lifts a portable

burner from the floor and places it onto the sink (executed on the desk in our real setup due to space constraints). During this process, the gas hose connecting the burner and the gas canister becomes detached. The second subtask requires the agent to press the emergency gas shutoff switch to stop further leakage. Since this step should be performed quickly to reduce the impact of the leak, the latency of code policy synthesis is critical. Our framework was able to generate and execute the necessary code with low latency, resulting in a faster response compared to baselines. In the final subtask, the agent reconnects the detached hose to the gas canister and toggles the emergency switch again to restore the gas flow, leaving the burner ready for use. This environment demonstrates how FCGRAFT's efficient code reuse and rapid synthesis enable timely adaptation to unexpected state changes, which in turn contributes to comparatively safer outcomes in practice. Figure 9 illustrates the entire execution sequence.

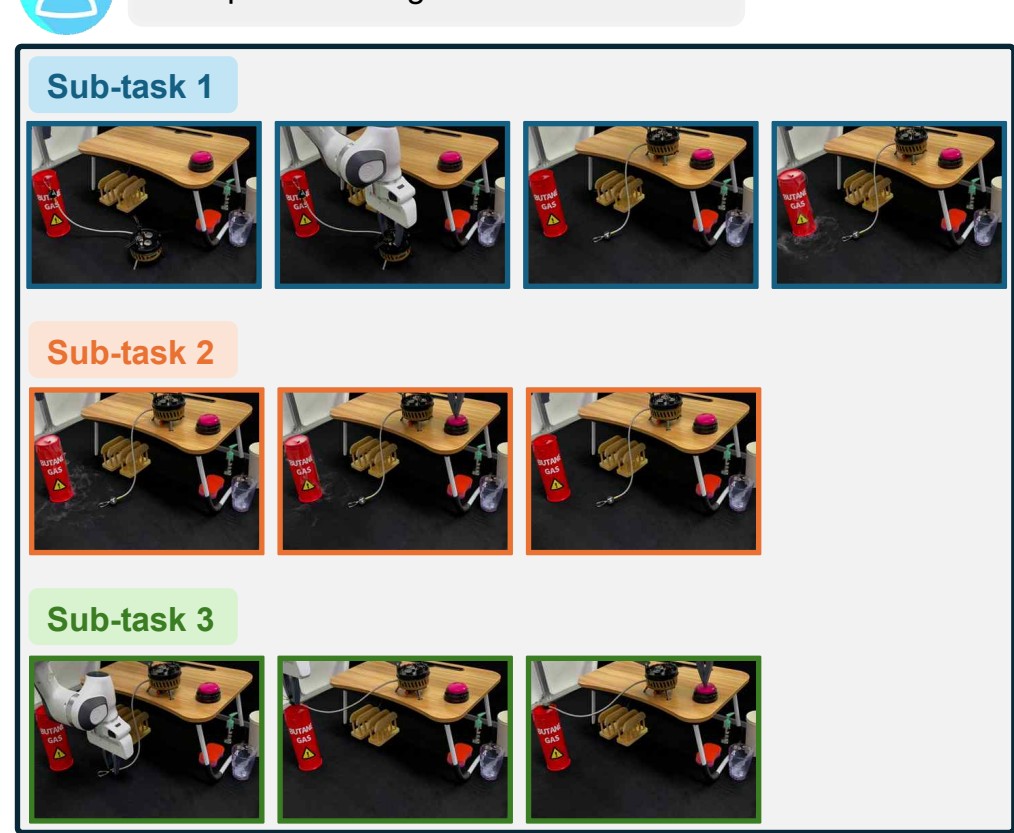

*Figure 9.* Real-world *Cooking Workstation Preparation*. The sequence involves three subtasks: (1) placing the burner onto the sink (desk in real setup), during which the gas hose disconnects; (2) pressing the emergency gas shutoff switch to quickly stop the leak; and (3) reconnecting the hose and toggling the switch to restore gas flow.

### C.3. Analysis on code cache warm-up

Figure 10 shows the bootstrapping performance of the code cache over a stream of 40 open-domain tasks. This experiment evaluates how the function-level KV caching performs when initialized with an empty cache and progressively populated through task execution. In Figure 10a, FCGRAFT consistently improves and maintains a higher SR, while Figure 10b shows a rapid reduction in PSL compared to other baselines. It also maintains a higher HR than RAGCache, as shown in Figure 10c, and achieves more efficient MU, as shown in Figure 10d, demonstrating the effectiveness of our code cache management strategy. These results indicate that FCGRAFT enables stable and efficient operation throughout the warm-up process, supporting consistent task performance via reliable bootstrapping within available resources.

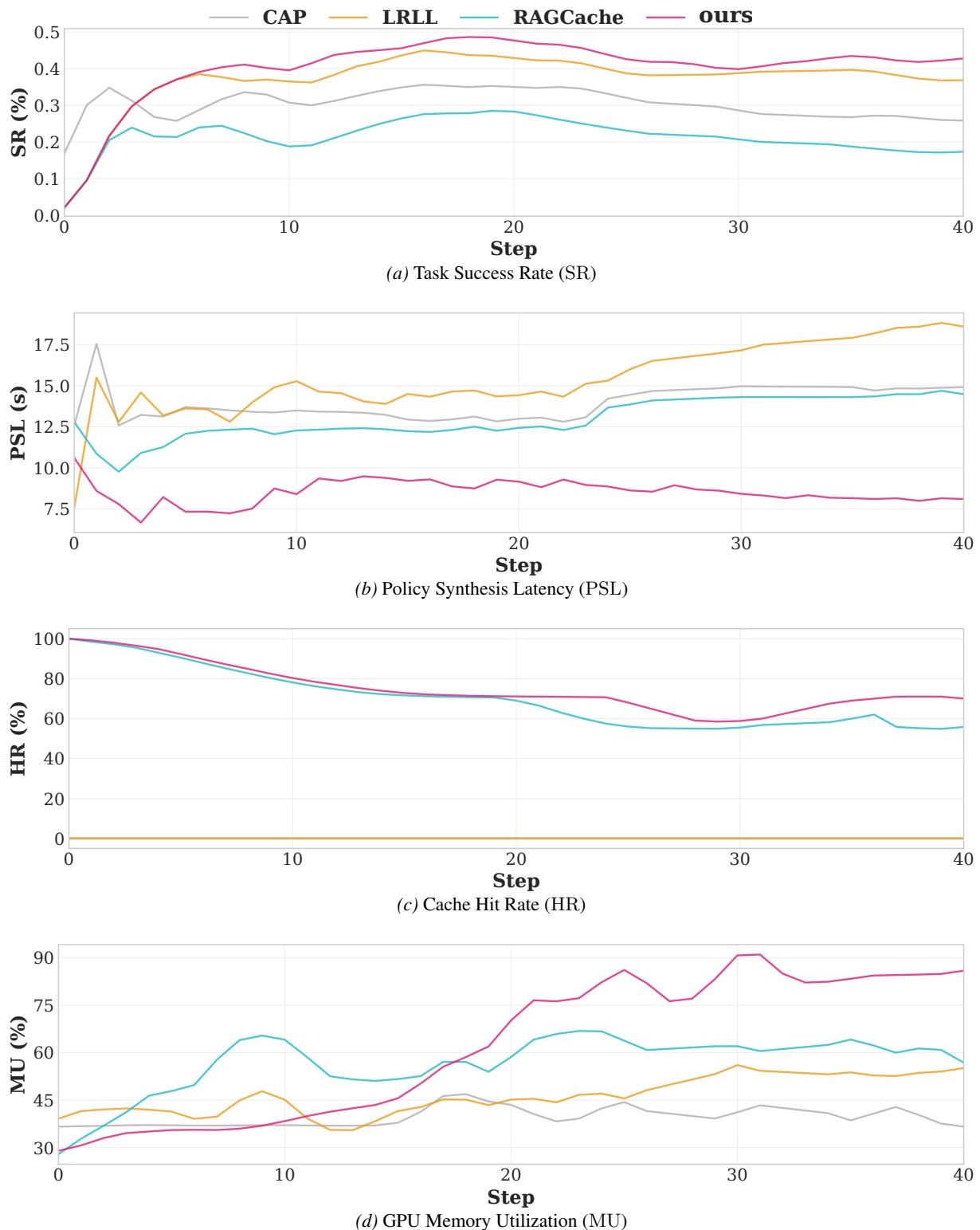

*(a)* Task Success Rate (SR)

*(b)* Policy Synthesis Latency (PSL)

*(c)* Cache Hit Rate (HR)

*(d)* GPU Memory Utilization (MU)

*Figure 10.* Analysis on code cache warm-up, with SR, PSL, HR, and MU over 40 tasks.

## C.4. Analysis on behavior consistency

Table 18 shows snapshot evaluations of the code cache at the *Initial*, *Middle*, and *Final* phases during a continual task stream of 25 tasks, providing the detailed results with all metrics corresponding to Table 3 in Section 5.3. At each phase, the entire task set is re-evaluated while preserving the current state of the code cache to assess the behavioral consistency of the accumulated code, reporting CSIM, FWT, BWT, and SR. The results show that FCGRAFT not only improves SR but also maintains consistent code structure across tasks; it achieves positive FWT across phases without forgetting (non-negative BWT). This demonstrates that FCGRAFT supports consistent behavior by transferring function-level knowledge in open-domain tasks.

*Table 18.* Continual adaptation performance on an open-domain scenario.

| RLBench | *Initial* (after task IDs 1–9) | | | | | | |
|---|---|---|---|---|---|---|---|
| Method | SR | FWT | BWT | PSL | NGT | HR | MU |
| CaP | $33.33 \pm 0.00$ | $0.00 \pm 0.00$ | $0.00 \pm 0.00$ | $9.34 \pm 0.06$ | $112.67 \pm 0.00$ | - | $24.86 \pm 0.00$ |
| LRLL | $33.33 \pm 0.00$ | $-6.25 \pm 0.00$ | $5.56 \pm 0.00$ | $18.42 \pm 0.42$ | $223.44 \pm 0.00$ | - | $25.09 \pm 0.00$ |
| RAGCache | $33.33 \pm 0.00$ | $0.00 \pm 0.00$ | $0.00 \pm 0.00$ | $9.76 \pm 0.18$ | $115.33 \pm 0.00$ | $22.00 \pm 0.00$ | $77.56 \pm 0.00$ |
| FCGRAFT | $\mathbf{44.44 \pm 0.00}$ | $2.08 \pm 0.00$ | $\mathbf{0.00 \pm 0.00}$ | $4.19 \pm 0.03$ | $222.50 \pm 0.71$ | $\mathbf{33.33 \pm 0.00}$ | $\mathbf{74.31 \pm 0.00}$ |

| RLBench | *Middle* (after task IDs 10–18) | | | | | | |
|---|---|---|---|---|---|---|---|
| Method | SR | FWT | BWT | PSL | NGT | HR | MU |
| CaP | $27.78 \pm 0.00$ | $0.00 \pm 0.00$ | $0.00 \pm 0.00$ | $9.40 \pm 0.12$ | $112.78 \pm 0.00$ | - | $16.58 \pm 0.00$ |
| LRLL | $33.33 \pm 0.00$ | $0.0 \pm 0.0$ | $5.56 \pm 0.00$ | $17.92 \pm 0.20$ | $217.11 \pm 0.00$ | - | $25.48 \pm 0.00$ |
| RAGCache | $33.33 \pm 0.00$ | $0.00 \pm 0.00$ | $0.00 \pm 0.00$ | $9.35 \pm 0.17$ | $111.78 \pm 0.00$ | $11.00 \pm 0.00$ | $77.47 \pm 0.00$ |
| FCGRAFT | $\mathbf{38.89 \pm 0.00}$ | $4.76 \pm 0.00$ | $\mathbf{0.00 \pm 0.00}$ | $3.36 \pm 0.13$ | $194.50 \pm 79.90$ | $\mathbf{41.67 \pm 3.92}$ | $\mathbf{74.26 \pm 2.30}$ |

| RLBench | *Final* (after task IDs 19–25) | | | | | | |
|---|---|---|---|---|---|---|---|
| Method | SR | FWT | BWT | PSL | NGT | HR | MU |
| CaP | $37.33 \pm 1.88$ | - | $0.00 \pm 0.00$ | $9.40 \pm 0.47$ | $112.95 \pm 4.01$ | - | $47.41 \pm 8.38$ |
| LRLL | $40.00 \pm 0.00$ | - | $4.00 \pm 0.00$ | $23.34 \pm 0.33$ | $274.22 \pm 0.00$ | - | $43.14 \pm 0.00$ |
| RAGCache | $36.44 \pm 0.00$ | - | $-2.00 \pm 2.83$ | $8.79 \pm 0.08$ | $105.22 \pm 0.00$ | $50.00 \pm 0.00$ | $77.60 \pm 0.00$ |
| FCGRAFT | $\mathbf{46.00 \pm 2.83}$ | - | $\mathbf{2.0 \pm 2.83}$ | $6.61 \pm 1.78$ | $188.50 \pm 60.10$ | $\mathbf{61.91 \pm 6.74}$ | $\mathbf{80.38 \pm 5.69}$ |

## C.5. Analysis on scalability under KV cache offloading

We measure KV cache transfer latency under realistic offloading scenarios across RLBench tasks. In FCGRAFT, high-locality interface entries are preferentially retained on GPU, while lower-scored code entries are offloaded to CPU memory and reloaded only when needed. GPU-to-CPU offloading can be scheduled outside the critical path of policy synthesis, whereas CPU-to-GPU reloads occur only when a required function cache is not resident on GPU. Therefore, the primary runtime overhead comes from selective reloads rather than routine offloading. Table 19 reports the measured transfer latency and frequency of CPU–GPU cache movement during validation.

The measured transfer latency is small relative to the average policy synthesis latency in our evaluated workloads. This is because most high-utility function KV entries remain resident on GPU, and only a small number of reuse-required entries need to be reloaded from CPU memory. However, this result should not be interpreted as a general claim that KV offloading overhead is always negligible. The overhead can become more significant for larger CodeLLMs, longer function implementations, larger batch sizes, or systems with lower CPU–GPU bandwidth. A more systematic characterization of storage hierarchy and interconnect bandwidth is left as future work.

## C.6. Analysis on the source of policy synthesis latency reduction

We decompose policy synthesis latency into time to first token (TTFT) and decoding time to isolate the efficiency gains of cache-stitching and cache-patching. Cache-stitching mainly affects the prefill stage because reused interface KV states remove the need to re-encode repeated function descriptions. Therefore, removing stitching substantially increases TTFT. In contrast, cache-patching mainly affects decoding because it regenerates only the localized faulty span instead of the entire

*Table 19.* Analysis on CPU–GPU offloading overhead for scalable KV cache reuse.

| Type | Latency (ms/synthesis) | Frequency (per validation) |
|------|------------------------|----------------------------|
| GPU $\to$ CPU | 20.33$\pm$2.62 | $\sim$ 20 |
| CPU $\to$ GPU | 23.42$\pm$3.19 | $\sim$ 4 |

policy. This decomposition clarifies that the PSL improvement reported in the main results arises from two complementary sources: reduced prefill through interface-level stitching and reduced decoding through localized code-level patching. Table 20 reports this latency decomposition under the full model and two ablated variants.

Removing cache-stitching increases TTFT by $5.3\times$, while additionally removing cache-patching mainly increases decoding time. This confirms that stitching primarily reduces prefill cost, whereas patching primarily reduces decoding cost.

*Table 20.* Analysis on the source of policy synthesis latency reduction.

| Method | TTFT (ms) | Decoding Time (s) |
|--------|-----------|-------------------|
| FCGRAFT | 215.10$\pm$4.30 | 3.37$\pm$0.84 |
| w/o. Cache-stitching | 1142.84$\pm$8.82 | 3.51$\pm$1.13 |
| w/o. Cache-stitching & patching | 1172.23$\pm$5.32 | 8.13$\pm$1.02 |

## C.7. Analysis on the practical validity of module-level cache reuse

This analysis evaluates the practical stability of module-level cache reuse, rather than proving strict token-level position independence of KV states. Because modern CodeLLMs use position-dependent attention mechanisms, naively reusing arbitrary token-level KV states may introduce positional artifacts. Our design avoids this setting by treating each function interface as a coherent reusable module and preserving its internal token order during composition. To isolate the effect of this reuse mechanism, we compare policies produced by cache-stitching against policies produced by full regeneration without cache reuse on the same task set. Cache-patching is disabled in both conditions, and all other settings, including task inputs, model checkpoints, prompts, and execution environments, are kept identical. The stitched variant composes policies from cached function-interface modules, whereas the full-regeneration variant generates the policy from the same instruction and retrieved context without KV reuse. Interpreter error rate measures syntactic and referential correctness of the generated code, while SR captures task-level correctness. Table 21 compares module-level cache reuse with full regeneration in terms of code validity, task success, and synthesis latency.

The results show that cache-stitching achieves comparable code correctness to full regeneration while substantially reducing policy synthesis latency. The comparable interpreter error rate indicates that reusing function-interface modules does not introduce additional syntactic or referential failures relative to full regeneration. At the same time, the lower PSL shows that the benefit comes from avoiding redundant prefill over reused interfaces. Thus, this result supports the practical validity of function-level module reuse, while not claiming strict token-level position independence.

*Table 21.* Analysis on the practical validity of module-level cache reuse.

| Method | Interpreter Error Rate | SR | PSL |
|--------|------------------------|-----|------|
| FCGRAFT w/ Stitching | 21.23 | 73.82 | 3.2s |
| Full regeneration | 22.04 | 71.53 | 5.7s |

## C.8. Analysis on the reliability of localized cache-patching

We analyze patching-triggered episodes to measure whether the target code span is correctly localized and whether the resulting patch resolves the failure. A patching attempt is counted as correctly localized when the selected code span

contains the function call, argument, object reference, or control branch that caused the observed failure signal. It is counted as incorrectly localized when the patch targets an unrelated span or modifies a surrounding region that does not explain the failure. Incorrect localization typically occurs when the failure signal is under-specified, when a perception-based postcondition mismatch is ambiguous, or when multiple functions jointly contribute to the observed failure. In such cases, patching may either fail to remove the triggering exception or modify a non-causal code region. Table 22 reports how localization correctness affects patching resolution.

Overall, 70% of patching attempts localize the correct target span. Successful recoveries arise primarily from correctly localized cases, while the incorrect-and-unresolved cases characterize a key failure mode of cache-patching. However, incorrect localization does not always imply task failure, because some patches still produce a behaviorally acceptable policy or partially recover the execution trajectory. For this reason, exception resolution should be interpreted as a stricter metric than end-to-end task success.

*Table 22.* Analysis on error localization and resolution in cache-patching.

|  | Resolved | Unresolved |
| --- | --- | --- |
| Correct localization | 43% | 27% |
| Incorrect localization | 16% | 14% |

### C.9. Analysis on the gap between exception resolution and task success

We further analyze the relationship between exception resolution and end-to-end task robustness. Although cache-patching improves execution robustness, environment-level and semantic exceptions are not always fully resolved after patching. This is expected because these exceptions often depend on perception feedback, physical execution outcomes, and postcondition checks, rather than only on syntactic or referential code validity. Therefore, the exception resolution rate should not be interpreted as equivalent to final task success.

In our evaluation, an exception is counted as resolved only when the triggered failure is correctly localized and the corresponding exception no longer reappears after patching. This is a stricter criterion than end-to-end task completion. For example, a patched policy may still complete the final task even if an intermediate postcondition mismatch remains unresolved, or if the feedback signal is conservative and triggers repair for a non-critical execution mismatch. This distinction is important because our unified exception interface is designed to favor sensitivity over specificity, capturing not only interpreter-level errors but also RobotError conditions and perception-driven postcondition mismatches.

Table 23 reports the end-to-end performance of resolved and unresolved patching cases. Resolved cases achieve 78.03% SR and 81.82% GC, while unresolved cases still achieve 52.41% SR and 55.82% GC. This indicates that unresolved exceptions do not necessarily imply catastrophic failure or final task failure. Instead, some unresolved patching attempts still partially correct execution or preserve enough valid structure for the task to complete.

*Table 23.* Analysis on the gap between exception resolution and task success.

| Resolution Status | Proportion | SR | GC |
| --- | --- | --- | --- |
| Resolved | 57.12% | 78.03% | 81.82% |
| Unresolved | 42.88% | 52.41% | 55.82% |

Table 24 compares FCGRAFT with full regeneration on the same subset of episodes in which patching was triggered. Replacing localized repair with full regeneration substantially reduces performance, achieving only 21.20% SR and 27.50% GC. This result shows that localized cache-patching provides a practical robustness benefit even when it does not fully remove the triggering exception. The benefit comes from preserving validated surrounding code and modifying only the affected span, rather than discarding the entire policy structure.

These results clarify the scope of our robustness claim. FCGRAFT does not guarantee that all environment-level or semantic failures are resolved, nor does it provide formal safety guarantees. Rather, it improves practical robustness by enabling

*Table 24.* Analysis on the benefit of localized repair under patching-triggered episodes.

| Method | SR | GC |
|---|---|---|
| FCGRAFT | 67.04% | 70.68% |
| Full regeneration | 21.20% | 27.50% |

localized repair from heterogeneous execution feedback and by preserving reusable validated code structures during recovery. Current performance on environment-level and semantic exceptions remains a limitation, especially when failure signals are ambiguous or perception feedback is incomplete.

## C.10. Ablation on model size and cache-patching

Table 25 provides the numerical values of the data plotted in Figure 11. It compares FCGRAFT across different CodeLLM sizes and cache-patching configurations. We evaluate the default setting (CoT-guided cache-patching) against two ablated variants: without CoT and with expert guidance replaced. As model size increases from 3B to 14B, FCGRAFT consistently achieves higher SR (26.57%, 44.50%, 53.75%) and GC (38.36%, 58.14%, 66.54%), while PSL ($2.90s$, $3.59s$, $4.53s$) also increases due to the larger model size. MU and HR remain relatively stable, indicating that FCGRAFT effectively balances performance and efficiency across model scales. For the ablation on cache-patching, using only execution feedback results in a noticeable drop in SR and GC, with slight reductions in PSL and NGT compared to the default setting. This highlights the efficiency of CoT-guided cache-patching. When replacing CoT with direct expert guidance, it achieves slightly higher SR and GC due to access to sufficient feedback, although the CoT-based approach remains highly competitive.

*Table 25.* Ablation on model size and cache-patching. Evaluation of CodeLLMs with different scales (3B, 7B, 14B), contrasting cache-patching against ablated settings: cache-patching without CoT (execution feedback only) and patching with direct human expert guidance.

| Setting | Model Size | SR | GC | PSL | NGT | HR | MU |
|---|---|---|---|---|---|---|---|
| | 3B | 24.67 | 35.54 | 2.71 | 121.14 | 66.27 | 49.71 |
| W/O GUIDANCE | 7B | 40.12 | 50.74 | 3.37 | 113.03 | 71.12 | 67.07 |
| | 14B | 45.80 | 60.63 | 4.48 | 103.00 | 74.50 | 82.63 |
| | 3B | 26.57 | 38.36 | 2.90 | 115.66 | 66.84 | 53.43 |
| COT GUIDANCE | 7B | 44.50 | 58.14 | 3.59 | 90.69 | 72.45 | 68.90 |
| | 14B | 53.75 | 66.54 | 4.53 | 104.96 | 74.56 | 84.86 |
| | 3B | 28.36 | 39.74 | 3.19 | 127.01 | 68.28 | 52.34 |
| EXPERT GUIDANCE | 7B | 46.69 | 59.15 | 3.89 | 123.15 | 71.96 | 68.97 |
| | 14B | 55.09 | 67.92 | 4.74 | 112.01 | 74.65 | 84.64 |

## C.11. Ablation on model choice

Table 26 extends Table 4 in Section 5.3 by reporting additional results, and presents the performance of FCGRAFT applied with four different LLM families, contrasting CodeLLMs with general-purpose LLMs. The results show that FCGRAFT equipped with CodeLLMs consistently outperforms its counterparts based on the LLMs, as well as CaP. On average, it achieves a $4.67\%$ higher SR, a $1.08\times$ reduction in PSL, and an $8.31\%$ higher HR compared to variants using the LLMs, while maintaining a more favorable trade-off than CaP. These findings demonstrate that FCGRAFT is not restricted to a specific LLM architecture and remains broadly compatible with diverse CodeLLMs.

## C.12. Ablation on FCGRAFT

Table 27 presents additional experimental results that complement Table 5 in Section 5.3, and analyzes the contribution of each component of FCGRAFT to overall performance. For the code cache structure, removing the Function-Interface tier $\mathcal{I}$ (denoted as w/o. $\mathcal{I}$), the Function-Code tier $\mathcal{C}$ (w/o. $\mathcal{C}$), or the entire cache $\mathcal{H}$ (w/o. $\mathcal{H}$) leads to a substantial drop in SR, confirming that the hierarchical design is crucial for reducing reasoning overhead and enabling reliable policy synthesis. In

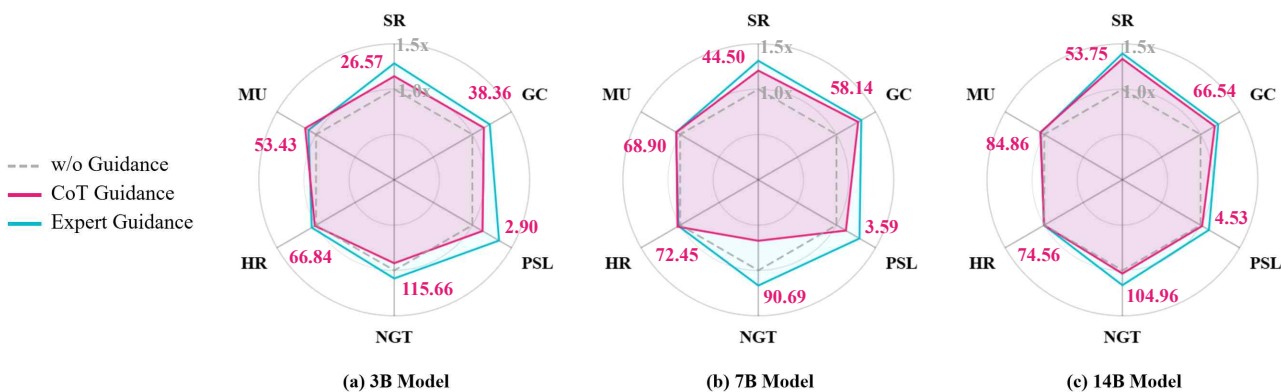

*Figure 11.* Ablation on model size and cache-patching. Evaluation of CodeLLMs with different scales (3B, 7B, 14B), contrasting cache-patching against ablated settings.

*Table 26.* Ablation on model choice.

| Family | Method | CodeLLM | SR | PSL | NGT | HR | MU |
|---|---|---|---|---|---|---|---|
| QWEN2.5 | FCGRAFT | ✔ | 29.62±0.01 | 2.89±0.41 | 106.67±1.70 | 58.63±1.28 | 70.40±2.30 |
| | FCGRAFT | ✘ | 26.38±0.23 | 3.12±0.43 | 158.67±1.25 | 47.12±3.94 | 68.03±2.97 |
| | CaP | ✔ | 23.63±0.16 | 7.94±0.47 | 110.32±3.40 | - | 12.93±0.49 |
| GEMMA | FCGRAFT | ✔ | 28.83±0.05 | 2.84±0.86 | 125.67±11.09 | 52.46±3.28 | 68.32±0.32 |
| | FCGRAFT | ✘ | 26.39±0.03 | 3.05±0.13 | 144.23±0.03 | 41.67±0.05 | 65.94±0.12 |
| | CaP | ✔ | 22.22±0.02 | 8.72±0.49 | 138.49±3.53 | - | 13.76±1.67 |
| LLAMA2 | FCGRAFT | ✔ | 31.12±1.32 | 3.32±0.09 | 151.67±3.77 | 57.91±4.27 | 67.99±0.92 |
| | FCGRAFT | ✘ | 25.33±1.07 | 3.46±0.26 | 168.03±4.01 | 52.85±1.29 | 62.03±0.88 |
| | CaP | ✔ | 17.83±2.46 | 11.48±0.23 | 156.40±3.49 | - | 11.20±0.30 |
| DEEPSEEK | FCGRAFT | ✔ | 30.23±5.71 | 2.33±0.34 | 130.00±17.34 | 67.23±3.28 | 71.09±2.03 |
| | FCGRAFT | ✘ | 23.04±3.21 | 2.64±0.86 | 134.00±1.73 | 61.34±3.42 | 68.03±2.83 |
| | CaP | ✔ | 18.98±0.41 | 13.07±1.13 | 146.00±3.88 | - | 13.28±0.24 |

particular, removing $\mathcal{C}$ (w/o. $\mathcal{C}$) disables direct code reuse, leading to invalid policy edits that yield the lowest SR despite the shortest PSL. Removing the semantic term in the locality score (w/o. $\ell_{\text{sema}}$) reduces long-horizon stability, indicating that semantic diversity in the code cache is critical for open-domain tasks. Replacing perplexity with a retriever ($\rightarrow$ Retrieval) yields comparable SR, but perplexity is preferable as it requires no additional encoder and achieves more efficient PSL and HR. Replacing cache-patching with full regeneration ($\rightarrow$ Regeneration) increases PSL and decreases SR, highlighting the efficiency of cache-patching in correcting errors with minimal decoding.

## C.13. Ablation on the contribution of each locality component

Table 28 presents an extended ablation designed to isolate the contribution of each locality component. For each setting, one component is assigned a weight of 1 while the remaining components are set to 0 (e.g., $(\alpha, \beta, \gamma) = (1, 0, 0)$). The evaluation is performed on open-domain task streams constructed to emphasize distinct structural properties, and three such task-stream types are considered in this analysis. **Repetitive task streams** consist of tasks in which similar operations appear repeatedly across episodes. In such settings, the dominant signal is the recurrence of individual functions, and prioritizing entries by invocation frequency is sufficient; accordingly, $\ell_{\text{freq}}$ alone captures the structure of these streams and yields the strongest performance. In contrast, **order-dependent task streams** require functions to be executed in specific sequences, and correctness depends on consistent ordering relations (e.g., inspect before transform). The key structure lies in conditional co-occurrence rather than raw frequency, making $\ell_{\text{asso}}$ the most effective component for maintaining useful function transitions. Finally, **semantically diverse task streams** include tasks with large functional variation or uncommon operations, where rare but semantically distinct functions become important. Preserving diversity in the cache is therefore

*Table 27.* Ablation on FCGRAFT. 'w/o.' denotes the removal of a component, and '→' indicates replacement with an alternative operation.

| Method | SR | PSL | NGT | HR | MU |
|---|---|---|---|---|---|
| FCGRAFT | 45.91±4.37 | 3.24±0.65 | 105.62±3.09 | 73.65±2.15 | 83.63±3.12 |
| w/o. $\mathcal{I}$ | 38.64±1.55 | 4.12±0.12 | 114.50±0.70 | 66.78±3.12 | 76.04±2.38 |
| w/o. $\mathcal{C}$ | 34.37±0.52 | 3.33±0.23 | 88.67±2.51 | 67.56±3.51 | 72.93±1.04 |
| w/o. $\mathcal{H}$ | 34.64±1.98 | 4.25±1.27 | 113.00±7.81 | - | 25.83±2.12 |
| w/o. $\ell_{\text{sema}}$ | 35.02±1.76 | 3.74±0.29 | 116.20±2.36 | 48.61±1.96 | 74.74±2.09 |
| → Retrieval | 41.92±0.72 | 4.42±0.48 | 117.39±3.67 | 43.06±2.45 | 46.72±2.49 |
| → Regenerate | 35.77±1.76 | 6.89±0.20 | 161.00±4.24 | 71.39±2.83 | 77.23±1.82 |

critical, and $\ell_{\text{sema}}$ alone best supports this requirement by prioritizing functions with high semantic uniqueness.

*Table 28.* Ablation of locality across heterogeneous task regimes.

| **RLBench** | *Repetitive* | | | *Order-dependent* | | | *Semantic diverse* | | |
|---|---|---|---|---|---|---|---|---|---|
| Methods | SR ($\uparrow$) | PSL ($\downarrow$) | HR ($\uparrow$) | SR ($\uparrow$) | PSL ($\downarrow$) | HR ($\uparrow$) | SR ($\uparrow$) | PSL ($\downarrow$) | HR ($\uparrow$) |
| $\alpha = 1$ | 36.43 | 3.34 | 67.04 | 34.34 | 3.32 | 61.04 | 35.12 | 3.40 | 60.23 |
| $\beta = 1$ | 33.32 | 3.43 | 56.04 | 37.20 | 3.33 | 69.32 | 33.92 | 3.92 | 54.95 |
| $\gamma = 1$ | 34.42 | 3.30 | 61.26 | 33.33 | 3.14 | 56.58 | 38.12 | 3.60 | 68.29 |

## C.14. Ablation on locality coefficient combinations

Table 29 extends the locality ablation by evaluating various coefficient combinations. While Table 28 isolates each component, this experiment explores how different weightings affect performance in practice. The configuration $(\alpha, \beta, \gamma) = (0.4, 0.3, 0.3)$ achieves the best SR and HR, indicating that a slightly higher weight on usage frequency $\ell_{\text{freq}}$ provides the most effective balance. Configurations that overly emphasize a single component (e.g., $\alpha = 0.6$ or $\beta = 0.6$) tend to degrade performance, confirming that all three components contribute meaningfully to cache management.

## C.15. Ablation of pre-population contents and cache-patching

Table 30 analyzes how pre-populated cache contents affect performance under KV-cache-enabled and KV-cache-disabled settings. The experiment is designed to show that the framework enables cache-patching of previously generated functions and supports low-latency synthesis without full regeneration. The purpose is not to indicate reliance on pre-population, but to demonstrate that even when the cache originates from a different domain, the system can efficiently modify and compose cached functions to achieve reliable execution in real-world conditions. To further analyze this capability, we conduct a controlled comparison between two types of stored code: (1) **program-level** code memory composed of monolithic code blocks, and (2) **function-level** code memory composed of decomposed, reusable functional units. For each type, we evaluate two variants-one that maintains the stored code in KV cache form and one that does not. All cached content originates from the simulation domain, and no real-world–specific information is preloaded. Across all four settings, the function-level memory with KV cache maintenance consistently shows higher task success, more stable execution behavior, and lower synthesis latency. These results indicate that function-level KV caching enables reliable code reuse, while cache-patching ensures fast adaptation, together supporting robust and efficient synthesis in open-domain.

*Table 29.* Ablation of locality coefficient combinations on RLBench.

| $\alpha$ | $\beta$ | $\gamma$ | SR ($\uparrow$) | PSL ($\downarrow$) | HR ($\uparrow$) |
|---|---|---|---|---|---|
| 0.4 | 0.3 | 0.3 | **45.81** | 3.32 | **74.03** |
| 0.3 | 0.4 | 0.3 | 43.56 | 3.28 | 67.36 |
| 0.3 | 0.3 | 0.4 | 43.44 | **3.38** | 68.20 |
| 0.5 | 0.3 | 0.2 | 44.92 | 3.42 | 69.45 |
| 0.3 | 0.5 | 0.2 | 42.45 | 3.38 | 66.71 |
| 0.2 | 0.3 | 0.5 | 43.39 | 3.37 | 67.88 |
| 0.5 | 0.2 | 0.3 | 45.18 | 3.39 | 71.82 |
| 0.3 | 0.2 | 0.5 | 43.82 | 3.40 | 67.38 |
| 0.2 | 0.5 | 0.3 | 42.43 | 3.35 | 66.11 |
| 0.6 | 0.2 | 0.2 | 42.54 | 3.38 | 67.36 |
| 0.2 | 0.6 | 0.2 | 41.20 | 3.42 | 63.36 |
| 0.2 | 0.2 | 0.6 | 41.94 | 3.34 | 63.36 |

*Table 30.* Ablation of pre-population content influence under KV cache enabled vs. KV cache disabled settings.

| **Real-world** | *Program-level* | | | | *Function-level* | | | |
|---|---|---|---|---|---|---|---|---|
| Methods | SR ($\uparrow$) | GC ($\uparrow$) | PSL ($\downarrow$) | HR ($\uparrow$) | SR ($\uparrow$) | GC ($\uparrow$) | PSL ($\downarrow$) | HR ($\uparrow$) |
| Non KV Cache | 44.44 | 53.33 | 8.04 | 74.50 | 77.78 | 77.78 | 6.21 | 87.25 |
| KV Cache | 66.67 | 71.11 | 4.52 | 74.50 | 88.89 | 88.89 | 3.80 | 87.25 |

## C.16. Ablation on model quantization.

Table 31 analyzes the effect of model quantization on FCGRAFT's performance and efficiency. Model quantization (Wolf et al., 2019) is largely complementary to our framework and can be applied alongside it to further improve efficiency. We additionally provide experimental results combining our method with quantization in the analysis below. These results further highlight that our framework pursues a different objective from conventional LLM serving while remaining fully compatible with it.

*Table 31.* Ablation of model quantization.

| **RLBench** | *Open-Composition* | | *Open-Perturbation* | | *Open-Evolution* | |
|---|---|---|---|---|---|---|
| Methods | SR ($\uparrow$) | PSL ($\downarrow$) | SR ($\uparrow$) | PSL ($\downarrow$) | SR ($\uparrow$) | PSL ($\downarrow$) |
| 4bit quantization | 40.32 | 2.53 | 36.05 | 2.58 | 30.33 | 3.21 |
| 8bit quantization | **45.91** | **3.24** | **42.82** | **3.47** | **33.98** | **4.39** |

