# OpenReview forum: "Functional Cache Grafting: Robust and Rapid Code-Policy Synthesis for Embodied Agents"
_ICML.cc/2026/Conference — ICML 2026 regular_

### Official Review · Reviewer_KgLT · 2026-03-11

**Soundness:** 3
**Presentation:** 3
**Significance:** 3
**Originality:** 3
**Overall Recommendation:** 4
**Confidence:** 3

**Summary:**

This paper introduces FCGRAFT, a function-level KV cache framework for CodeLLM-based embodied agents operating under the Code-as-Policies paradigm. The key idea is that robotic skill programs are composed of reusable function calls, so rather than regenerating or re-encoding entire programs from scratch, the system maintains a two-tier cache consisting of Interface segments (I, capturing function signatures and type annotations) and Function-Code segments (C, capturing full implementations). Two composition operations are defined: cache-stitching assembles cached I-segments to form a compact function context for new code generation, while cache-patching swaps in updated C-segments when functions need correction at runtime. The system uses Qwen2.5-Coder-14B as its backbone and is evaluated on ALFRED, TEACh, and RLBench in simulation, plus a real-world Franka Emika Research 3 setup. Across nine baselines spanning CodeLLM programming methods, memory-based approaches, and KV cache systems, FCGRAFT achieves the highest success rate in all tested scenarios, including three open-domain settings (Open-Composition, Open-Perturbation, Open-Evolution) designed to test generalization and adaptation.

**Compliance With Llm Reviewing Policy:**

Affirmed.

**Key Questions For Authors:**

1. What mechanism ensures position-independence when composing KV cache segments from different original positions? Qwen2.5-Coder uses RoPE, so naively stitching segments should introduce position-dependent artifacts. Do you re-encode, adjust position indices, or rely on a specific property of the Interface segments that makes them robust to position shifts?

2. MU is reported in the appendix tables but absent from the main Tables 1 and 2. Can you include memory utilization for FCGRAFT and the KV cache baselines (CAG, EPIC, PromptCache) in the main results?

3. How accurate is the error localization for cache-patching? What fraction of patching operations target the correct function, and what happens when the system patches the wrong cache entry?

4. Can you clarify the status of Eq. 1 - is it an operational objective that guides cache construction, or a conceptual framing? The locality score weights (α, β, γ in Eq. 3) appear to be fixed heuristics rather than learned.

5. The real-world evaluation uses 9 trials per environment. Can you expand this, or at least report confidence intervals? The sample size is small relative to the claims being made.

6. For the continual learning metrics (FWT, BWT), how is task ordering handled? Are results averaged over multiple orderings, or is a single fixed sequence used?

**Limitations:**

Section 5.1 acknowledges that FCGRAFT maintains isolated per-agent caches, which may limit scalability in multi-agent settings, and discusses future work on cache sharing and learning-based eviction. This is reasonable but misses several issues that affect the current claims. The position-independence mechanism, which is foundational to the stitching operation, is not described in enough detail to assess its robustness or failure modes. The real-world evaluation is small (9 trials per environment across two scenarios) and the high variance in Table 2 reflects this, yet the paper does not flag the limited scale as a limitation. MU being relegated to appendix tables rather than the main results leaves a gap in the efficiency narrative. The error localization component lacks any characterization of when it fails, which matters because incorrect patching could degrade rather than improve execution. And three seeds without significance testing means some of the reported improvements may not be reliable, particularly in scenarios where the SR gap between methods is modest.

**Strengths And Weaknesses:**

**Strengths:**

The function-level caching abstraction is genuinely novel in the embodied AI context. Prior KV cache work (PromptCache, EPIC, CAG) operates at the token or prompt level without any awareness of code structure, and prior CaP work treats programs as monolithic text. Decomposing cached representations along function boundaries - separating interface from implementation - is a natural fit for how robotic skill libraries are actually organized, and it enables the stitching/patching operations that make the framework useful. I think the two-tier design is well motivated: stitching gives the LLM a compact function-level context for generation, while patching allows targeted runtime correction without invalidating the rest of the cache. These address different failure modes (missing context vs. buggy execution) and the ablation in Table 5 confirms both matter, with removing either tier degrading SR noticeably.

The experimental coverage is also strong. Nine baselines across three categories, three simulation benchmarks with meaningfully different characteristics, a real-world robot evaluation, and three open-domain scenarios that go beyond standard benchmark splits. The continual learning metrics (Forward Transfer, Backward Transfer) borrowed from that literature are a useful addition for evaluating how well cached functions generalize to new tasks without degrading old ones. The model-family ablation in Table 4 across Qwen2.5, Gemma, Llama2, and DeepSeek shows FCGRAFT's gains are not an artifact of one particular CodeLLM, and the component ablation in Table 5 gives confidence that the individual design choices (Interface tier, Code tier, semantic component of the locality score) are each pulling their weight.

**Weaknesses:**

My main concern is the position-independence assumption underlying cache-stitching. The paper claims that Interface segments are "position-agnostic" and can be freely composed regardless of their original positions in the encoding sequence, citing prior work (Yao et al., 2025; Hu et al., 2024; Gim et al., 2024). But what mechanism actually ensures this? Qwen2.5-Coder uses RoPE, where KV states encode absolute position information, so naively concatenating segments from different positions should introduce artifacts. CacheBlend and EPIC are cited but the paper does not specify whether it adopts any particular re-encoding or position-adjustment strategy from those works. This is not a minor implementation detail - it is the technical foundation that makes stitching work, and it needs to be described explicitly rather than deferred to citations.

The error localization mechanism for cache-patching is also underspecified. Section 3.2 describes identifying erroneous code spans and using a CoT trace (Wei et al., 2022) to locate the root cause, and Figure 3 shows runtime tracebacks being used in practice. But the paper does not report how reliable this localization is - no accuracy numbers or false-positive rates for the patching trigger. Since incorrect patching could corrupt a working cache entry, understanding the failure modes here matters. Relatedly, the paper claims that "stitching eliminates internal errors," which is a strong statement that holds only when the cached functions are themselves correct and the position-independence assumption is satisfied.

There are some gaps in the reported results that weaken parts of the narrative. The paper defines a Memory Utilization (MU) metric in Section 4.1 but only reports it in appendix tables, omitting it from the main-body Tables 1 and 2 - for a framework whose pitch is efficient caching, surfacing the actual memory footprint alongside SR and PSL in the main results would strengthen the story. Eq. 1 frames the objective as jointly optimizing SR, PSL, and CSIM with weighting factors η and μ, but this reads as a conceptual framing rather than an operational objective, and the cache management scoring (Eq. 3, with α, β, γ) uses fixed heuristic weights rather than learned ones. This is fine but should be stated explicitly. The real-world evaluation, while welcome, involves only 9 trials per environment (N=3 per task across 3 tasks, as described in the appendix), and the high variance in Table 2 reflects this. Three seeds with no significance testing across the simulation experiments is also thin, especially for scenarios like RLBench Open-Evolution where the gap between FCGRAFT and the next-best method (LRLL) is only about 1.5 SR points.

---

> ### Author Rebuttal · Authors · 2026-03-31
>
> Dear reviewer KgLT,
>
> Thank you for raising this important point.
>
> > **[W1 & Q1]**
>
> The “position-agnostic” is not meant in the strict token-level sense, but in a practical module-level sense. Following PromptCache [1], we treat the reusable unit as a function-level KV segment whose internal positional structure is preserved during reuse. Thus, reuse does not require full re-encoding as long as the segment remains internally coherent. For more details, please refer to our response to W1 of Reviewer `s7RR`.
>
> > **[W2 & Q3]**
>
> To assess the reliability of the error-localization step used for cache-patching, we analyzed cases that triggered patching after an exception. For each case, we evaluated whether the correct code span was localized and whether the resulting patch successfully restored execution.
>
> We find that 70% of patching attempts correctly localize the target span. Among all attempts, 43% are resolved and 27% remain unresolved. The remaining 30% are incorrectly localized, of which 16% are still resolved and 14% remain unresolved.
>
> Overall, these results suggest that the localization step is effective in practice, with successful recoveries arising primarily from correctly localized cases.
>
> ||Resolved|Unresolved|
> |-|-|-|
> |Correct Localization|43%|27%|
> |Incorrect Localization|16%|14%|
>
> We further analyze exception-triggered patching by exception type. Interpreter-level exceptions account for about 21.2% of cases and have the resolution rate at 75.8%, while environment-level exceptions are more frequent but harder to resolve, with a resolution rate of 53.4%.
>
> |Exception Type|Frequency|Resolution Rate|
> |-|-|-|
> |Interpreter|21.2%|75.4%|
> |Environment|78.8%|53.4%|
>
> > **[W3 & Q4]**
>
> Eq. 1 is a conceptual framing of the desired trade-off, η and μ are not learned parameters. The trade-off is realized procedurally through cache-stitching, cache-patching, and cache management.
> Likewise, the locality-score weights α, β and γ in Eq. 3 are heuristic coefficients, not learned parameters. In the current implementation, we set them to (0.4,0.3,0.3), following the sensitivity analysis reported in Appendix D.9.
>
> > **[W3 & Q5]**
>
> In our real-world evaluation, the 9 trials for each real-world task scenario are not simple repetitions of same setup. Rather, they reflect variations in task composition and instruction, together with randomized object locations and workspace configurations. Specifically, for each scenario, we construct 9 distinct trial conditions and evaluate each condition over 3 randomized runs, resulting in 27 trials in total per scenario.
> To address the reviewer’s concern, we report 95% confidence intervals for the real-world SR using the Wilson score interval.
>
> |Methods|SR|95% CI|SR|95% CI|
> |-|-|-|-|-|
> ||_Office Desk Rearrangement_||_Cooking Workstation Preparation_||
> |CaP|33.3±0.0|[18.6, 52.2]|51.9±6.4|[34.0, 69.3]|
> |LRLL|55.6±11.1|[37.3, 72.4]|55.6±0.0|[37.3, 72.4]|
> |RAGCache|44.4±22.2|[27.6, 62.7]|37.0±6.42|[21.5, 55.8]|
> |Ours|77.8±11.1|[59.2, 89.4]|81.5±12.8|[63.3, 91.8]|
>
> These intervals show that, although the sample size is modest, the performance advantage of FCGraft remains substantial in both real-world scenarios.
>
> > **[W3 & Q6]**
>
> For Table 3 (FWT and BWT), we use a single fixed task ordering to isolate the effect of function-level KV caching on consistent behavior, as randomizing order would conflate task-sequencing effects with cache accumulation [2].
>
> For the main simulation results in Table 1, three seeds across three distinct open-domain scenarios already provide substantial variation. Rather than randomizing task order, we keep the sequence fixed and evaluate across diverse scenarios to maintain a controlled protocol while still testing robustness under different open-domain conditions. The three seeds are also not simple repetitions, as each induces different initial environment configurations, including object placement, object subset selection, and spatial layout.
>
> Regarding RLBench Open-Evolution, where the SR gap between FCGraft and LRLL is modest, our objective is not to optimize SR alone, but to jointly optimize SR and PSL while maintaining behavioral consistency. Accordingly, the results should also be interpreted through the Rank metric in Table 1, which captures the effectiveness-efficiency trade-off. Under this metric, FCGraft still maintains a margin of 0.19 over LRLL, while the actual second-best method is CAG with a gap of 0.14. Detailed results are provided in Appendix.
>
>
> > **[W1 & W3 & Q2]**
>
> In the revision, we will clarify that our method does not assume strict position independence of cached KV states, add MU columns to Tables 1 and 2 for FCGraft and the KV cache baselines (CAG, EPIC, PromptCache, and RAGCache), and further clarify the real-world evaluation protocol described in Appendix B.4, with explicit MU comparisons against baselines.
>
> ---
>
> [1] Prompt Cache: Modular Attention Reuse for Low-Latency Inference
>
> [2] Gradient episodic memory for continual learning

---

> > ### Author Rebuttal · Reviewer_KgLT · 2026-04-03
> >
> > The error localization breakdown (70% correct, 59% overall resolution) was what I needed. The 14% incorrect+unresolved rate gives a clear picture of when patching fails. The interpreter vs environment exception split is a useful detail.
> >
> > Real-world evaluation being 27 trials per scenario (9 conditions x 3 runs) is more substantial than the "9 trials" I read in the paper. The Wilson CIs confirm the gap holds.
> >
> > Eq. 1 and Eq. 3 being conceptual/heuristic with sensitivity analysis in D.9 is clear. On position-independence, dropping the "position-agnostic" framing and clarifying the actual mechanism in the revision is the right move. Table 4 across four model families provides enough practical evidence.

---

> > > ### Author Response · Authors · 2026-04-04
> > >
> > > We sincerely thank the reviewer for the thoughtful engagement and for confirming that the concerns have been adequately addressed. We are pleased that our rebuttal has fully resolved the raised points. Your detailed feedback has been invaluable in strengthening both the clarity and rigor of the paper.
> > >
> > > Sincerely,
> > >
> > > The Authors

---

### Official Review · Reviewer_evMB · 2026-03-12

**Soundness:** 3
**Presentation:** 4
**Significance:** 3
**Originality:** 3
**Overall Recommendation:** 5
**Confidence:** 3

**Summary:**

The paper proposes FCGraft (Functional Cache Grafting), a framework designed to resolve the high latency and low robustness issues inherent in the Code-as-Policies (CaP) paradigm for embodied agents. FCGraft maintains a two-tier library of validated code implementations and their corresponding Transformer Key-Value (KV) caches. During policy synthesis, the framework employs "cache-stitching" to compose executable policies directly from cached function states, eliminating redundant prefill computation. If execution fails, "cache-patching" utilizes execution feedback and Chain-of-Thought (CoT) to locally regenerate only the erroneous code spans, reusing the prefix KV cache. The authors evaluate FCGraft across three simulation benchmarks (ALFRED, TEACh, RLBench) and real-world robotic tasks, demonstrating significant improvements in synthesis latency and task success rates.

**Compliance With Llm Reviewing Policy:**

Affirmed.

**Key Questions For Authors:**

1. The cache-patching mechanism is triggered by exceptions raised during execution. How does FCGraft handle "silent failures" where the generated API sequence is syntactically correct and executes without raising an exception, but fails to achieve the visual/physical goal due to external disturbances?

2. Regarding the cache management scheme, offloading low-scored entries to DRAM is mentioned. In long-horizon tasks where the cache grows significantly, what is the specific latency overhead introduced by swapping KV caches back and forth between DRAM and GPU?

3. In the Appendix, Time to First Token (TTFT) is listed as a metric , but the main results primarily report Policy Synthesis Latency (PSL). Could you provide a brief breakdown of TTFT versus the actual decoding time to better isolate the exact efficiency gains provided strictly by the KV cache stitching?

**Limitations:**

yes

**Strengths And Weaknesses:**

Strengths:
- Presentation: As noted during the initial reading, the paper is exceptionally well-structured and visually clear. Figures 1 and 2 excellently illustrate the contrast with conventional CaP and the internal mechanisms of the two-tier cache.
- Soundness (Empirical Evaluation): The empirical validation is extensive. Evaluating across three diverse simulators and validating on a physical 7-DoF Franka Emika robot under open-domain scenarios strongly supports the claims of robustness. The ablation studies, particularly regarding different LLM families (Qwen2.5, Gemma, Llama2, DeepSeek), provide excellent depth.
- Significance & Originality: Applying KV caching—a technique typically reserved for system-level LLM serving acceleration—at the function level specifically for robotic policy synthesis is a highly practical and creative cross-pollination of ideas. It addresses a genuine bottleneck (latency) in deploying LLM agents in real-time environments.

Weaknesses:
- Soundness (Error Handling Limits): The cache-patching mechanism appears heavily reliant on explicit, well-defined execution feedback, such as interpreter-level exceptions (e.g., NameError) or strict environment-level APIs (RobotError). It is somewhat unclear how the framework handles "silent" perception failures where the code executes flawlessly but the physical action fails due to unobserved real-world physics.
- Soundness (Scalability overhead): While the locality score manages cache eviction effectively , storing function-level KV caches on GPU memory poses a strict upper bound. In a truly lifelong deployment, the latency overhead of frequently offloading/loading KV states between GPU and DRAM is not deeply quantified in the main text.

---

> ### Author Rebuttal · Authors · 2026-03-31
>
> Dear reviewer evMB,
>
> Thank you for raising this important point.
>
> > **[W1 & Q1] Handling silent failures beyond interpreter-level exceptions**
>
> In our setting, silent failures that do not raise interpreter-level exceptions can be categorized into:
>
> (1) Physical execution failures, where execution completes but the physical action fails (e.g., unsuccessful grasp) and
>
> (2) Postcondition violations, where the action is executed, but the desired goal state (e.g., drawer opened) is not achieved.
>
> Such semantic failures may arise without explicit interpreter-level exceptions.
>
> As described in Appendix C.1 and Table 11, our feedback system includes interpreter-level exceptions such as SyntaxError, ClassNotFound, and FileNotFoundError, as well as embodied execution failures represented through RobotError [1]. In real-world deployment, such failures includes both (1) physical execution failures and (2) postcondition violations, which are converted into structured failure signals through perception-based checks. We verify execution outcomes using depth cameras, object detectors, and a VLM-based state check. For example, when the observed state does not match the expected postcondition, this mismatch is converted into a defined failure signal. Therefore, cache-patching is not limited to interpreter-level exceptions, but can also be triggered by the embodied failure signals, including both (1) and (2).
>
> FCGRAFT focuses on utilizing heterogeneous feedback through a unified interface for localized repair, not on designing perception modules, and thus scales naturally with feedback quality.
>
> > **[W2 & Q2] Latency overhead of KV cache offloading in lifelong deployment**
>
> The table below reports KV cache transfer latency under realistic offloading scenarios across three RLBench benchmarks.
>
> |Type|Latency (ms/synthesis)|Frequency (per validation)|
> |-|-|-|
> |GPU → CPU|20.33±2.62|~20|
> |CPU → GPU|23.42±3.19|~4|
>
> Importantly, in FCGraft’s cache management scheme (Section 3.1), low-scored entries, particularly those in the Function-Code tier $\mathcal{C}$, are offloaded to DRAM, while the Function-Interface tier $\mathcal{I}$ is preferentially retained on GPU.
> Offloading (GPU → CPU) is performed during task execution rather than during policy synthesis, and therefore does not lie on the critical path of PSL.
> In contrast, reloading (CPU → GPU) is required before reuse and constitutes the primary source of potential latency overhead.
>
> This frequency asymmetry indicates that most offloaded entries remain in DRAM, with only reuse-required entries selectively reloaded. These reload times account for less than 1% of the average PSL, and therefore do not materially affect end-to-end performance. FCGraft maintains an average HR of approximately 70%, meaning that most high-utility function KV entries remain resident on the GPU, and only cache misses require reloading from DRAM.
>
> Beyond the scenarios above, we identify the following conditions as the primary factors that could make reload overhead significant:
>
> 1. when unusually long function implementations lead to large per-entry KV sizes.
> 2. when scaling to larger CodeLLMs, where KV size grows with layer count, head count, and head dimension, while interconnect bandwidth remains fixed.
>
> While our current results suggest that reuse dominates transfer overhead in realistic continual-task settings, we agree that systematically characterizing this trade-off at larger scales is an important direction for future work.
>
> > **[Q3] Breakdown of TTFT vs. decoding time within PSL**
>
> While TTFT is listed as a metric in the Appendix, the main results report PSL, which is defined as the average end-to-end time to produce an executable code policy and therefore include both the prefill phase and the decoding phase.
>
>
> Cache-stitching and cache-patching target these two phases differently: stitching reduces prefill by reusing cached KV states $\mathcal{I_{KV}}$, while patching reduces decoding by generating only the erroneous span rather than the full code.
>
> More specifically, cache-stitching reuses $\mathcal{I_{KV}}$ to reduce TTFT, whereas cache-patching reuses $x_{pre}^{KV}$ to reduce decoding time. Since patching is invoked only after an exception is detected, its contribution to the overall TTFT metric is comparatively limited.
>
> As shown below, removing cache-stitching increases TTFT by 5.3×, while additionally removing cache-patching increases decoding time by 2.3× with minimal TTFT change, confirming that the two mechanisms target distinct latency components.
>
> | |TTFT (ms)|Decoding time (s)|
> |-|-|-|
> |Ours|215.10±4.30|3.37±0.84|
> |w/o. Cache-stitching|1142.84±8.82|3.51±1.13|
> |w/o. Cache-stitching & patching|1172.23±5.32|8.13±1.02|
>
> ---
>
> [1] Inner monologue: Embodied reasoning through planning with language models

---

### Official Review · Reviewer_s7RR · 2026-03-12

**Soundness:** 3
**Presentation:** 3
**Significance:** 3
**Originality:** 3
**Overall Recommendation:** 4
**Confidence:** 3

**Summary:**

This paper aims to addresss the latency and robustness limitations of code-as-policies for embodied agents. It designs a framework that maintains function-level kv caches and generates new code policies by stitching and patching cached function segments.

**Compliance With Llm Reviewing Policy:**

Affirmed.

**Final Justification:**

My main concern of this paper lies on the novelty side: the paper seems to apply many existing tricks/methods from the code generation domain to the embodied code-as-policy domain, without much innovation on code-to-embodied agent itself. But considering the technical soundness, results, and the authors' efforts and explanations in the rebuttal, I'd like to raise my score to 4 (though 3.5 would more precisely reflect my assessment).

**Key Questions For Authors:**

See weakness section.

**Limitations:**

yes

**Strengths And Weaknesses:**

Strengths:

1. This paper is well motivated, as it addresses the latency and robustness issues of embodied agents, which are critical needs in many real-world scenarios.

Weaknesses:

1. The cache stitching relies on KV states being "position-agnostic," but I see no technical explanation or ablation provided in the main content to support this critical assumption.
2. The core techniques (two-tier cache, KV reuse, locality eviction, FIM repair) seem to be generic code generation methods. Is there any design specifically addresses challenges inherent to embodied agents, such as partial observability, long-horizon decision making, and safety?
3. In real-world deployments, errors often translate to unsafe operations. Given that the framework relies on a try-except mechanism for cache patching, it is unclear how safety can be guaranteed when such exceptions are triggered in practice.
4. While I'm unfamiliar with kv cache for code generation or skill composition in this line of work, I think there may be prior work addressing similar challenges to those presented in this paper, excluding the embodied agent aspects. I hope the authors can expand the related work section to discuss such work more thoroughly and explicitly highlight the differences and novelty of their approach.

---

> ### Author Rebuttal · Authors · 2026-03-31
>
> Dear reviewer s7RR,
>
> We thank the reviewer for raising this concern.
>
> > [W1] Position-agnostic assumption lacks technical justification
>
> Here, ‘position-agnostic’ is used in a practical module-level sense rather than in the strict token-level sense.
> Our core premise follows the modular reuse view of PromptCache [1]. When the reusable unit is a coherent module, effective KV reuse can remain feasible as long as the module preserves its internal positional relationships and is composed with appropriately assigned position IDs.
> EPIC [2] further indicates that, in such non-prefix reuse, the key challenge is often boundary-level inter-segment attention rather than the need to recompute the entire cached segment.
>
> Specifically, the practical stability of interface-level KV reuse in our implementation is supported by the following considerations.
> Following PromptCache, our method treats the reusable unit as a function-level KV cache, which preserves the positional relationships of tokens within each function.
> Following EPIC, which shows that non-prefix KV reuse can be maintained by applying minimal recomputation to boundary tokens that may otherwise become attention sinks, we also consider this at function-level KV cache boundaries, as a complementary step, to mitigate inter-function interference and stabilize cache-stitching in practice.
>
> > [W3] How FCGraft handles unsafe operations beyond interpreter-level exceptions
>
> In normal real-world deployments, errors can translate into unsafe operations in two cases:
>
> (i) when the world state unexpectedly changes before the current primitive skill is completed, causing the skill to become inappropriate during execution, or
>
> (ii) when the robot remains in a hazardous state for an extended period without timely corrective action.
>
> We agree that relying only on interpreter-level errors is insufficient for safe cache-patching in real-world deployments.
>
> As described in Appendix C.1 and Table 11, our feedback system includes interpreter-level errors such as SyntaxError, ClassNotFound, and FileNotFoundError, as well as environment-level errors and postcondition violations represented as RobotError [3]. Accordingly, FCGraft extends exception triggering beyond built-in interpreter-level exceptions to environment-level failures and perception-based feedback, so that cache-patching can be invoked immediately for both cases (i) and (ii). This table demonstrates how effectively FCGraft detects and resolves each category of silent failure in real-world deployment.
>
> |Exception Type|Frequency|Resolution Rate|
> |-|-|-|
> |Interpreter|21.23%|75.37%|
> |Environment|41.76%|52.03%|
> |Semantic|37.01%|55.01%|
>
> While we do not claim that FCGraft provides safety guarantees, it offers a practical safety benefit by enabling rapid corrective code-policy synthesis once unexpected situations are detected. FCGraft does not improve the perception modules themselves, but unifies heterogeneous failure signals into a common interface for localized code repair, and naturally scales as perception and postcondition checking improve.
>
> > [W2 & W4] Novelty and distinction from prior works
>
> Our core contribution is not KV reuse in isolation, but its reformulation into a function-level cache-grafting framework for CodeLLM-based embodied policy synthesis in open-domain environments.
>
> - **Partial observability and dynamic environment changes.** FCGraft combines observation-conditioned stitching with execution-coupled patching to repair only the affected region on instruction-observation mismatch or execution-time state changes, rather than regenerating the entire policy.
> - **Long-horizon and continual embodied tasks.** FCGraft treats function-level cache entries as reusable skill-like units, recomposing recurring substructures via its two-tier code cache.
> - **Safety.** Rather than formal guarantees, our design improves practical robustness by reusing validated function implementations to reduce internal control errors during stitching, and localizing runtime failures through patching.
>
> The novelty therefore lies in reformulating KV cache reuse into a skill-based framework for embodied code-policy synthesis, with function-level skill accumulation and localized time-sensitive adaptation.
>
> While we included an expanded related work discussion in Appendix A, we agree that the distinction from prior work should be made clearer in the main paper. In the revision, we will explicitly clarify how FCGraft differs from prior code-as-policies, memory-based methods, and KV-caching approaches, especially in its function-level cache abstraction and execution-coupled repair design.
>
> ---
>
> [1] Prompt Cache: Modular Attention Reuse for Low-Latency Inference
>
> [2] EPIC: Efficient Position-Independent Caching for Serving Large Language Models
>
> [3] Inner monologue: Embodied reasoning through planning with language models

---

> > ### Author Rebuttal · Reviewer_s7RR · 2026-04-03
> >
> > 1. While the authors refer to PromptCache/EPIC for the module-level position-agnostic assumption, it would be good to directly ablate FCGraft (e.g., stitched vs. fully regenerated code correctness).
> > 2. The reported resolution rates of ~52–55% for environment and semantic errors mean nearly half of real-world failures go unresolved, which raises my concern about the practical robustness claim.

---

> > > ### Author Response · Authors · 2026-04-06
> > >
> > > Dear s7RR,
> > >
> > > We thank the reviewer for the thoughtful follow-up questions, which allow us to provide further clarification and additional experimental evidence.
> > >
> > > > Additional Q1: Direct ablation of cache-stitching versus full regeneration on code correctness
> > >
> > > We additionally conducted a direct ablation comparing code produced by cache-stitching against code produced by full regeneration without cache reuse on the same task set.
> > > To isolate the effect of stitching itself, cache-patching was disabled in both conditions, and all other settings were kept identical.
> > > We report interpreter error rate, task success rate (SR), and policy synthesis latency (PSL).
> > > Here, interpreter error rate measures syntactic and referential correctness of the generated code, while SR captures task-level correctness.
> > >
> > > |Method|Interpreter error rate|SR|PSL|
> > > |-|-|-|-|
> > > |FCGraft|21.23%|73.82%|3.2s|
> > > |Full regeneration|22.04%|71.53%|5.7s|
> > >
> > > The results show that cache-stitching achieves comparable code correctness to full regeneration, both in interpreter-level validity and task success, while substantially reducing policy synthesis latency.
> > > This provides direct evidence within FCGraft that cache-stitching does not degrade code correctness in practice, and supports the module-level position-agnostic assumption underlying our design.
> > >
> > > > Additional Q2: Practical robustness under low exception resolution rates
> > >
> > > We agree that the reported 52–55% resolution rates for environment-level and semantic exceptions indicate that cache-patching still fails to fully eliminate a substantial fraction of such exceptions. We will clarify this limitation more explicitly in the revision. At the same time, we would like to emphasize that this resolution metric should not be interpreted as equivalent to overall practical robustness or final task success.
> > > In our previous response, an exception is counted as “resolved” only when the triggered failure is correctly localized and the corresponding exception no longer reappears after patching, which is a stricter criterion than whether the overall task eventually succeeds. This distinction is consistent with our evaluation protocol, where SR measures end-to-end task completion and GC measures sub-goal completion.
> > >
> > > This gap is especially important because our unified exception interface is intentionally broad. As described in Appendix C.1 and Table 11, the framework not only captures interpreter-level failures, but also a wide range of RobotError conditions and perception-driven postcondition mismatches.
> > > These signals are designed to favor sensitivity over specificity: the system may trigger repair even for non-critical skill failures or conservative postcondition mismatches. This includes cases where the final task goal is already achieved but the observed state does not exactly match the expected intermediate state. As a result, an “unresolved” exception does not necessarily imply catastrophic failure or even final task failure.
> > >
> > > |Resolution status|Proportion|SR|GC|
> > > |-|-|-|-|
> > > |Resolved|57.12%|78.03%|81.82%|
> > > |Unresolved|42.88%|52.41%|55.82%|
> > >
> > > To make this point concrete, we conducted an additional analysis over the subset of episodes in which patching was triggered, as shown in the resolution breakdown table above. We found that resolved cases achieved 78.03% SR and 81.82% GC, while unresolved cases still achieved 52.41% SR and 55.82% GC.
> > >
> > > By contrast, on the same patching-triggered subset, replacing the repair opportunity with full regeneration from scratch yielded only 21.20% SR and 27.50% GC, as shown in the overall comparison table below. This shows that the exception-resolution metric is substantially stricter than end-to-end task success, and that localized cache-patching still provides a meaningful robustness benefit even when it does not fully eliminate the triggering exception. In other words, unresolved patching attempts are often still useful because they can partially correct execution and preserve enough valid structure for the task to complete.
> > >
> > > |Method|SR|GC|
> > > |-|-|-|
> > > |FCGraft|67.04%|70.68%|
> > > |Full regeneration|21.20%|27.50%|
> > >
> > > Therefore, our claim is not that FCGraft resolves most real-world failures, nor that it provides formal robustness or safety guarantees. Rather, our claim is that localized cache-patching offers a practical robustness advantage over full regeneration by recovering a substantial fraction of failures with much lower decoding latency, and by improving final task success even in cases where the triggering exception is not fully removed. We will revise the paper to make this distinction between exception resolution and end-to-end robustness explicit, and to state more clearly that current performance on environment and semantic exceptions remains a limitation.

---

### Official Review · Reviewer_SkmG · 2026-03-14

**Soundness:** 3
**Presentation:** 4
**Significance:** 3
**Originality:** 3
**Overall Recommendation:** 5
**Confidence:** 3

**Summary:**

This paper focuses on improving the high latency problem in current methods that generate code as policies for embodied agents. The paper identifies that the latency comes from two factors: (1) redundant prefill computation and (2) the need to regenerate policies from scratch every time. To address this problem, the authors propose the FCGRAFT framework, which leverages a two-tier code cache consisting of cached function interfaces and cached full function implementations. They also introduce mechanisms to make caching efficient: (1) cache-stitching, which concatenates KV states of function segments to compose new policies, and (2) cache-patching, which regenerates only the faulty code span using prefix KV states, avoiding full program regeneration. The authors evaluate the method on several embodied benchmarks (including ALFRED, TEACh, and RLBench) as well as real-world robotic manipulation tasks, reporting improvements in both task success rate and policy synthesis latency compared to several baseline approaches.

**Compliance With Llm Reviewing Policy:**

Affirmed.

**Final Justification:**

My only weakness has been fully resolved by the authors’ response. Therefore, I am updating my score.

**Key Questions For Authors:**

Please see the weaknesses above.

**Limitations:**

yes

**Strengths And Weaknesses:**

## Strengths
- The paper is nicely written. The writing is clear and easy to follow, and the figures are clean and intuitive.
- The cache-stitching and cache-patching mechanisms are clean ways to improve the efficiency of program synthesis.
- Strong empirical evaluation across multiple benchmarks shows improvements in both robustness and efficiency.
- The paper includes detailed ablation studies and analyses, such as behavioral consistency evaluation and model ablations across different LLM families.

## Weaknesses
- Unclear contribution of KV caching to success rate. The ablation in Table 5 shows that replacing cache-stitching with retrieval reduces the success rate. However, it is not immediately clear why KV caching would significantly affect task success, as KV reuse primarily reduces redundant prefill computation and should mainly impact latency. It would be helpful if the authors clarified why KV caching also leads to higher success rate in this setting.

---

> ### Author Rebuttal · Authors · 2026-03-31
>
> Dear reviewer SkmG,
>
> We thank the reviewer for raising this important point. We would like to clarify in FCGraft, function-level KV caching plays a fundamentally different role beyond reducing redundant prefill computation.
>
> > [W1] Contribution of cache-stitching to success rate, beyond prefill acceleration
>
> In FCGraft, KV caching is repurposed not only to accelerate prefill but also to reuse previously validated function-level code structures encoded as attention states. As a result, policy synthesis is biased away from unconstrained token-by-token generation and toward a more structure-aware composition process, in which the CodeLLM is guided by reusable validated functions rather than generating an entire policy from scratch.
>
> Specifically, by enabling function-level code reuse through KV caching, FCGraft improves success rate through three related effects:
> (1) reducing the effective search space, (2) preventing many internal structural errors before they occur, and (3) improving behavioral consistency across tasks.
>
> - For (1), injecting cached KV states biases decoding toward previously validated program structures, which reduces the likelihood of invalid API compositions or inconsistent control flows.
> - For (2), cache-stitching reuses validated function structures, so many structural errors that would otherwise arise in newly synthesized policies are avoided upfront. Cache-patching then efficiently corrects the remaining localized mismatches without requiring full regeneration.
> - For (3), function-level KV reuse keeps policy synthesis closer to previously validated code structures, yielding more stable and consistent behavior across related tasks in open-domain environments.
>
> These effects are empirically supported by the table below, excerpted from Appendix Table 15, where FCGraft outperforms naive robotic programming methods, memory-based approaches, and token-level KV caching baselines in success rate.
>
> |Method|SR|PSL|NGT|HR|MU|
> |-|-|-|-|-|-|
> |CaP|31.23±3.32|11.39±0.53|123.43±3.55|-|23.08±0.88|
> |HELPER|35.42±1.32|11.48±0.23|154.18±1.45|-|26.48±1.93|
> |RAGCache|27.26±1.08|9.09±0.63|149.35±5.02|59.02±1.03|75.34±4.83|
> |Ours|45.91±4.37|3.24±0.65|105.62±3.09|73.65±2.15|83.63±3.12|
>
> Specifically, general CodeLLM-based programming methods such as CaP serve as a reference for unconstrained full-policy generation, thereby highlighting the benefit of (1) search space reduction in FCGraft.
> Memory-based approaches reuse textual artifacts, but they do not directly benefit from function-level KV stitching or cache-patching, making them less effective at (2) preventing structural errors upfront and correcting localized errors efficiently.
> Token-level KV caching baselines reuse attention states for acceleration, but they do not reuse previously validated function-level structures, and therefore do not offer the same robustness and (3) behavioral consistency benefits as FCGraft.
>
> This interpretation is further supported by the behavioral consistency analysis in Table 3 of our paper, where FCGraft maintains higher structural consistency and positive forward transfer (FWT) without forgetting, as reflected by non-negative backward transfer (BWT).
> These results confirm that FCGraft’s gains arise from function-level reuse and localized correction, rather than from prefill acceleration alone.
>
> **→ Retrieval variant.**
> We would also like to clarify a potential misunderstanding regarding Table 5.
> The variant labeled ${→ Retrieval}$ does not replace cache-stitching with retrieval. Instead, it replaces our perplexity-based semantic scoring $\ell_{sema}$with retriever-based semantic matching using sentence-embedding cosine similarity. The observed difference in success rate comes from how semantic relevance is scored, rather than from removing KV caching itself. In practice, the perplexity-based design used in FCGraft is preferable because it requires no additional retrieval encoder and achieves a better efficiency trade-off, with stronger HR and lower PSL.
>
> **→ Regeneration variant.**
> Additionally, replacing cache-patching with full regeneration (denoted as ${→ Regeneration}$ ) degrades performance because it removes the ability to perform localized error correction, increasing decoding overhead as reflected in the PSL values above.
>
> The table below is reproduced from Table 5 in the main text for reference, presenting the ablation study results.
>
> |Method|SR|PSL|HR|MU|
> |-|-|-|-|-|
> |Ours|45.91±4.37|3.24±0.65|73.65±2.15|83.63±3.12|
> |→ Retrieval|41.92±0.72|4.42±0.48|43.06±2.45|46.72±2.49|
> |→ Regenerate|35.77±1.76|6.89±0.20|71.39±2.83|77.23±1.82|
>
> Finally, we acknowledge that this causal connection between KV reuse and success rate was not sufficiently emphasized in the main table discussion, although it is partially described in the Appendix. We will revise the paper to make this connection clearer.

---

> > ### Author Rebuttal · Reviewer_SkmG · 2026-04-02
> >
> > I thank the authors for their thorough response. My concerns are fully resolved and I have updated my score.

---

> > > ### Author Response · Authors · 2026-04-03
> > >
> > > We sincerely thank the reviewer for the constructive and detailed comments on our paper. The main concerns raised will be thoroughly addressed in the upcoming revision. The feedback has been very helpful in improving the clarity and quality of the paper.
> > >
> > > Best regards,
> > >
> > > The Authors

---

### Decision · Program_Chairs · 2026-04-30

**Decision:**

Accept (regular)

**Comment:**

This paper addresses the latency and brittleness of LLM-generated robot code. It introduces a novel mechanism for caching previous successful program structures, and to use that to bias the generation of new programs, leveraging ideas similar to KV caching.

Overall the approach is technically sound and novel, and the empirical results are strong, demonstrating both an increase in robustness and decrease in latency. The detailed reviews make recommendations for specific revisions, which the authors should address in the final version.